# T-REGS: Minimum Spanning Tree Regularization for Self-Supervised Learning

Julie Mordacq[1,2]    David Loiseaux[1,2]    Vicky Kalogeiton[2]    Steve Oudot[1,2]

[1] Inria Saclay    [2] LIX, CNRS, École Polytechnique, IP Paris

## Abstract

Self-supervised learning (SSL) has emerged as a powerful paradigm for learning representations without labeled data, often by enforcing invariance to input transformations such as rotations or blurring. Recent studies have highlighted two pivotal properties for effective representations: *(i)* avoiding *dimensional collapse*-where the learned features occupy only a low-dimensional subspace, and *(ii)* enhancing *uniformity* of the induced distribution. In this work, we introduce T-REGS, a simple regularization framework for SSL based on the length of the Minimum Spanning Tree (MST) over the learned representation. We provide theoretical analysis demonstrating that T-REGS simultaneously mitigates dimensional collapse and promotes distribution uniformity on arbitrary compact Riemannian manifolds. Several experiments on synthetic data and on classical SSL benchmarks validate the effectiveness of our approach at enhancing representation quality. Code is available here.

## 1   Introduction

Self-supervised learning (SSL) has emerged as a powerful paradigm for learning meaningful data representations without relying on human annotations. Recent advances, particularly in visual domains [4, 31, 17, 55, 60], have demonstrated that self-supervised representations can rival or even surpass those learned through supervised methods. A dominant approach in this field is *joint embedding self-supervised learning* (JE-SSL) [12, 4, 56], where two networks are trained to produce similar embeddings for different views of the same image (see Figure 1). The fundamental challenge in JE-SSL is to prevent *representation collapse*, where networks output identical and non-informative vectors regardless of the input. To address this challenge, researchers have developed various strategies. *Contrastive approaches* [12, 30] encourage embeddings of different views of the same image to be similar while pushing away embeddings of different images. *Non-contrastive methods* bypass the need of negative pairs, often employing *asymmetric architectures* [13, 28, 10] or enforcing decorrelation among embeddings through *redundancy reduction* [4, 65, 63].

Recent studies have identified a more subtle form of collapse known as *dimensional collapse* [33, 35, 29, 41]. This phenomenon occurs when the embeddings span only a lower-dimensional subspace of the representation space, leading to high feature correlations and reduced representational diversity. Such a collapse can significantly impair the model's ability to capture the full complexity of the data, limiting performance on downstream tasks [26]. Another crucial aspect of representation quality is *uniformity*, which measures how evenly the embeddings are distributed across the representation space. It ensures that the learned representations preserve the maximum amount of information from the input data and avoid clustering in specific regions of the space. This property is fundamental because it helps maintain the discriminative power of the representations, and allows for better generalization to downstream tasks [23, 61, 52, 24].

39th Conference on Neural Information Processing Systems (NeurIPS 2025).

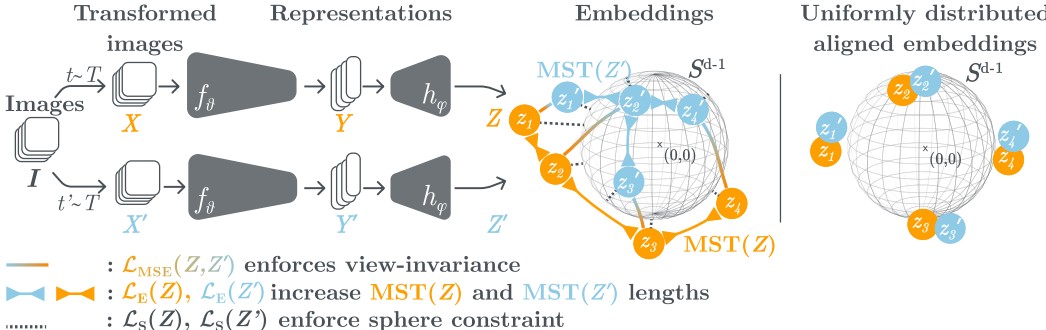

Figure 1: **Overview of T-REGS.** *(Left)* Two augmented views $X, X'$ are encoded by $f_\theta$ and projected by $h_\phi$ into embeddings $Z, Z'$. Training jointly: *(i)* minimizes the Mean Squared Error, $\mathcal{L}_{\mathrm{MSE}}(Z, Z')$, to enforce view invariance (or alternatively the objective function of a given SSL method, $\mathcal{L}_{\mathrm{SSL}}(Z, Z')$, when used as an auxiliary term); *(ii)* maximizes the minimum-spanning-tree length on each branch, $\mathcal{L}_{\mathrm{E}}(Z)$ and $\mathcal{L}_{\mathrm{E}}(Z')$, repelling edge-connected points in $\mathrm{MST}(Z)$ and $\mathrm{MST}(Z')$; and *(iii)* applies sphere constraints $\mathcal{L}_{\mathrm{S}}(Z)$ and $\mathcal{L}_{\mathrm{S}}(Z')$. *(Right)* As a result, T-REGS induces uniformly distributed embeddings without dimensional collapse.

While existing methods have made progress in mitigating dimensional collapse and enforcing uniformity, they present limitations: *contrastive methods* are sensitive to the number of negative samples [26, 4], and require large batch sizes, which can be computationally expensive; *redundancy reduction* methods, which enforce the covariance matrix to be close to the identity matrix, only leverage the second moment of the data distribution and are blind to, e.g., concentration points of the density, which can prevent convergence to the uniform density (Figure 6); and *asymmetric methods* lack theoretical grounding to explain how the asymmetric network helps prevent collapse [63].

Given these limitations, Fang et al. [23] suggested rethinking the notion of good SSL regularization, and proposed an Optimal Transport-based metric that satisfies a set of four principled properties (*instance permutation*, *instance cloning*, *feature cloning*, and *feature baby constraints*) that prevent dimensional collapse and promote sample uniformity. Their approach has its drawbacks: the optimal transport distances are costly to compute in general, and the proposed closed formula for accelerating the computation holds only on the sphere and requires square roots over SVD computations, which may lead to numerical instabilities.

To address these limitations, we propose T-REG, a novel regularization approach that is conceptually simple, easy to implement, and computationally efficient. T-REG naturally satisfies the four principled properties of Fang et al. [23] (Appendix G) and provably prevents dimensional collapse while promoting sample uniformity (Figure 2a). These properties make T-REG suitable for joint-embedding self-supervised learning, where it can be applied independently to each branch, yielding T-REGS (see Figure 1).

The central idea of T-REG is to maximize the length of the minimum spanning tree (MST) of the samples in the embedding space. It has strong theoretical connections to the line of work on statistical dimension estimation via entropy maximization [57] (Section 4.1). More explicitly, given a point cloud $Z$ in Euclidean space, a spanning tree (ST) of $Z$ is an undirected graph $G = (V, E)$ with vertex set $V = Z$ and edge set $E \subset V \times V$ such that $G$ is connected without cycle. We define the length of $G$ as:

$$E(G) := \sum_{(z,z')\in E} \|z - z'\|_2. \tag{1}$$

A minimum spanning tree of $Z$, denoted by $\mathrm{MST}(Z)$, is an ST of $Z$ that minimizes length $E$; it is unique under a genericity condition on $Z$. Since the length of $\mathrm{MST}(Z)$ scales under rescaling of $Z$, maximizing it alone leads the points to diverge (see Figure 2b). To prevent trivial scaling, T-REG constrains embeddings to a compact manifold, encouraging full use of the representation dimension and a uniform distribution.

Our main contributions can be summarized as follows:

*i)* We introduce T-REG (Equation (6)), a regularization technique that maximizes the length of the minimum spanning tree (MST) while constraining embeddings to lie on a hypersphere (Section 4).

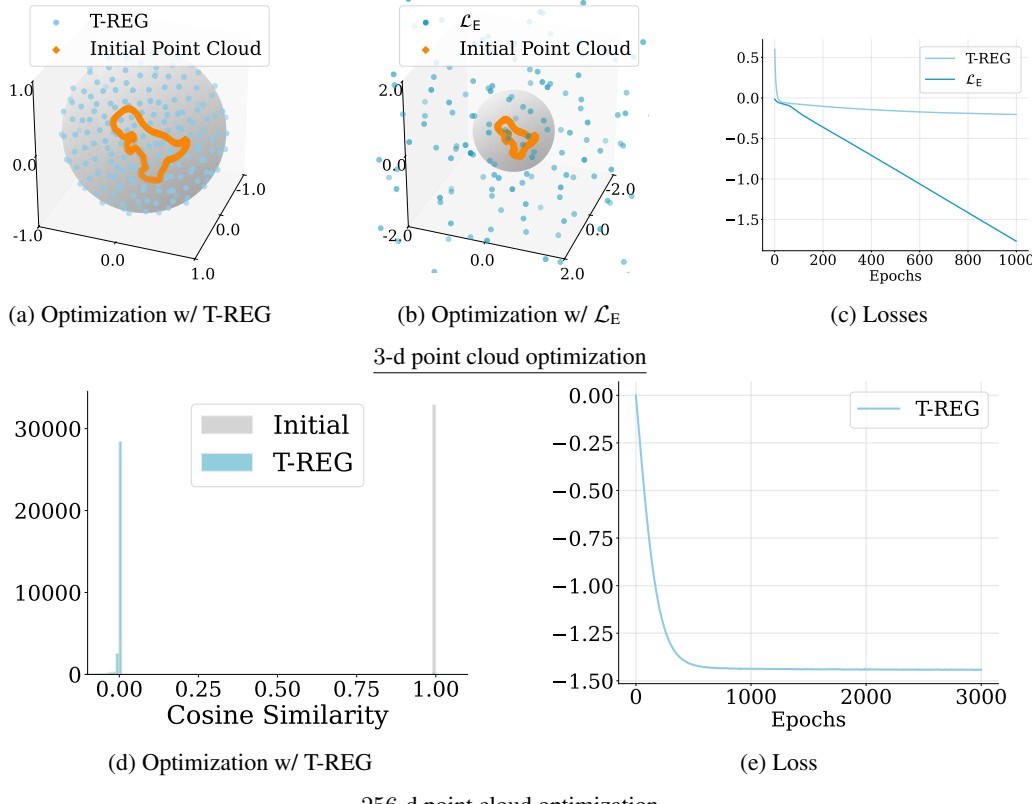

Figure 2: **Illustration of T-REG with synthetic data.** *(a-c) 3-d point cloud analysis:* (a) T-REG successfully spreads points uniformly on the sphere by combining MST length maximization and sphere constraint, (b) using only MST length maximization leads to excessive dilation, (c) stable convergence of T-REG whereas $\mathcal{L}_E$ alone fails to converge. *(d-e) Higher-dimensional analysis (256-d):* (d) T-REG enforces effective convergence to the 255-d regular simplex (Theorem 4.1), (e) stable optimization behavior of T-REG.

    *ii)* We show both theoretically and empirically that T-REG naturally prevents dimensional collapse while enforcing sample uniformity (Sections 4.1 and 4.2).

    *iii)* We apply T-REG to SSL either as standalone regularization, combining directly with view-invariance, or as an auxiliary loss to existing methods, yielding the T-REGS framework—whose effectiveness is evaluated through experiments on standard JE-SSL benchmarks (Section 5).

## 2 Related Work

Our work builds upon recent advances in Joint-Embedding Self-Supervised Learning (JE-SSL), which can be broadly categorized into two main approaches: contrastive and non-contrastive methods [26].

*(i) Contrastive methods* [32, 44, 12, 30, 14, 9] are commonly based on the InfoNCE loss [46]. These methods encourage the embeddings of augmented views of the same image to be similar, while ensuring that embeddings from different images remain distinct. Contrastive pairs can either be sampled from a memory bank, as in MoCo [30, 14], or generated within the current batch, as in SimCLR [12]. Furthermore, *clustering-based methods* [8, 9, 27] can also be seen as contrastive methods between prototypes, or clusters, instead of samples. SwAV [9], for instance, learns online clusters using the Sinkhorn-Knopp transform. However, despite their effectiveness, both approaches require numerous negative comparisons to work well, which can lead to high memory consumption. This limitation has spurred the exploration of alternative methods.

*(ii) Non-contrastive methods* bypass the reliance on explicit negative samples. *Distillation-based methods* incorporate architectural strategies inspired by knowledge distillation to avoid representation

collapse, such as an additional predictor [13], self-distillation [10], or a moving average branch as in BYOL [28]. Meanwhile, *redundancy reduction methods* [65, 20, 4, 68, 63, 56, 62] attempt to produce embedding variables that are decorrelated from each other, thus avoiding collapse. These methods can be broadly categorized into two groups: those that enforce soft whitening through regularization and those that perform hard whitening through explicit transformations. BarlowTwins [65] and VICReg [4] regularize the off-diagonal terms of the covariance matrix of the embedding to have a covariance matrix that is close to the identity. W-MSE [20] transforms embeddings into the eigenspace of their covariance matrix (batch whitening) and enforces decorrelation among the resulting vectors. CW-REG [62] introduces channel whitening, while Zero-CL [68] combines both batch and channel whitening techniques. Building upon these methods, INTL [63] proposed to modulate the embedding spectrum and explore functions beyond whitening to prevent dimensional collapse.

In this paper, we introduce a self-supervised learning (SSL) approach that uses a novel regularization criterion: the maximization of the embeddings' minimum spanning tree (MST) length. Our approach aligns with the framework proposed by Fang et al. [23], satisfying the same four principled properties: *instance permutation*, *instance cloning*, *feature cloning*, and *feature baby constraints*.

## 3 MST **and dimension estimation**

Steele [57] studies the total length of a minimal spanning tree (MST) for random subsets of Euclidean spaces. Let $X_n$ be an i.i.d.[1] $n$-sample drawn from a probability measure $P_X$ with compact support on $\mathbb{R}^d$. For $d \geq 2$, Theorem 1 of [57] controls the growth rate of the length of $\mathrm{MST}(X_n)$ as follows:

$$E\left(\mathrm{MST}(X_n)\right) \sim Cn^{(d-1)/d} \text{ almost surely, as } n \to \infty, \tag{2}$$

where $\sim$ denotes asymptotic convergence, and where $C$ is a constant depending only on $P_X$ and $d$.

This asymptotic rate makes it possible to derive several estimators of the intrinsic dimension of the support of a measure of its samples [53, 2, 15]. Among these estimators, the following one theoretically coincides with the usual dimension in non-singular cases, and it empirically coincides with the real-valued Hausdorff dimension [21] in classical manifold examples or even fractal examples, such as the Cantor set or the Sierpiński triangle.

**Definition 3.1.** Given a bounded metric space $M$, the MST dimension of $M$, denoted by $\dim_{\mathrm{MST}}(M)$, is the infimal exponent $d \in \mathbb{N}$ such that $E\left(\mathrm{MST}(X)\right)/|X|^{\frac{d-1}{d}}$ is uniformly bounded for all finite subsets $X \subseteq M$:

$$\dim_{\mathrm{MST}}(M) := \inf\{d : \exists C \text{ such that } E\left(\mathrm{MST}(X)\right)/|X|^{\frac{d-1}{d}} \leq C \text{ for every finite subset } X \text{ of } M\}.$$

**Persistent Homology Dimension.** The MST also appears in Topological Data Analysis (TDA) [47], where it relates to the *total persistence in degree* $0$ *of the Rips filtration* [47]. Moreover, Persistent Homology (PH) has been used to define a family of fractal dimensions [1], $\dim_{\mathrm{PH}}^i(M)$, for each homological degree $i \geq 0$. In particular, for $i = 0$ this coincides with the MST-based dimension, i.e., $\dim_{\mathrm{PH}}^0(M) = \dim_{\mathrm{MST}}(M)$. The PH dimension can be derived from entropy computations and has already been used in several dimension-estimation applications [58, 5, 19]. In this work, we directly leverage the connection to the entropy to obtain uniformity properties on compact Riemannian manifolds (see Section 4.1.2).

MST **optimization.** TDA further provides a mathematical framework for optimizing the length of $\mathrm{MST}(X)$ with respect to the point positions of $X$ [11, 40]. Within this framework, $E\left(\mathrm{MST}(X)\right)$ is differentiable almost everywhere, with derivatives given by the following simple formula:

$$\forall x \in X, \quad \nabla_x E\left(\mathrm{MST}(X)\right) = \sum_{\substack{(x,z) \text{ edge} \\ \text{of } \mathrm{MST}(X)}} \nabla_x \|x - z\|_2 = \sum_{\substack{(x,z) \text{ edge} \\ \text{of } \mathrm{MST}(X)}} \|x - z\|_2^{-1} (x - z). \tag{3}$$

Furthermore, under standard assumptions on the learning rate, stochastic gradient descent is guaranteed to converge almost surely to critical points of the functional. In particular, Equation (3) shows that each pair of points forming an edge in the MST exerts a repulsive force on the other during optimization.

---

[1]Independent and identically distributed.

MST **computation.** Given a finite point set $X \subset \mathbb{R}^d$, several classic sequential procedures exist to compute $\text{MST}(X)$, notably Kruskal's, Prim's, and Boruvka's algorithms, which all have at least quadratic running time in the size of $X$. However, fast GPU-based parallelized implementations exist, for instance [22], which unifies Kruskal's and Boruvka's algorithms and incorporates optimizations such as path compression and edge-centric operations.

## 4  T-REG: Minimum Spanning Tree based Regularization

Our regularization T-REG has two terms: a length-maximization loss $\mathcal{L}_E$ that decreases with the length of the minimum spanning tree, and a soft sphere-constraint $\mathcal{L}_S$ that increases with the distance to a fixed sphere $\mathbb{S}$. These two terms combined force the embeddings to lie on $\mathbb{S}$ (or close to it), while spreading them out along $\mathbb{S}$.

Formally, given $Z = \{z_1, ..., z_n\} \subseteq \mathbb{R}^d$, the MST length maximization loss is defined as:

$$\mathcal{L}_E(Z) = -\frac{1}{n} E\left(\text{MST}(Z)\right), \tag{4}$$

where $E\left(\text{MST}(Z)\right)$ denotes the length of the MST of $Z$. The soft sphere-constraint is given by:

$$\mathcal{L}_S(Z) = \frac{1}{n} \sum_i (\|z_i\|_2 - 1)^2. \tag{5}$$

It penalizes points that move away from the unit sphere. Maximizing the MST length alone would cause the points to diverge to infinity; the sphere constraint prevents this by keeping the embeddings within a fixed region around $\mathbb{S}$ (Figures 2b and 2c). The overall T-REG loss combines these two terms:

$$\mathcal{L}_{\text{T-REG}}(Z) = \gamma\, \mathcal{L}_E(Z) + \lambda\, \mathcal{L}_S(Z), \tag{6}$$

where $\gamma$ and $\lambda$ are hyperparameters controlling the trade-off between spreading out the embeddings and maintaining them on the sphere.

The remainder of the section provides a theoretical analysis (Section 4.1) and empirical evaluation (Section 4.2) of T-REG.

### 4.1  Theoretical analysis

#### 4.1.1  Behavior on small samples

We begin by considering the case where $n \leq d + 1$. It is particularly relevant since, in SSL, batch sizes are often smaller than or comparable to the ambient dimension. In order to account for the effect of the soft sphere constraint, we assume the points of $X$ lie inside some fixed closed Euclidean $d$-ball $B$ of radius $r$ centered at the origin (see below for the explanation).

**Theorem 4.1.** *Under the above conditions, the maximum of $E\left(\text{MST}(X)\right)$ over the point sets $X \subset B$ of fixed cardinality $n$ is attained when the points of $X$ lie on the sphere $S = \partial B$, at the vertices of a regular $(n-1)$-simplex that has $S$ as its smallest circumscribing sphere.*

Recall that a $k$-*simplex* is the convex hull of a set of $k + 1$ points that are affinely independent in $\mathbb{R}^d$—which is possible only for $k \leq d$. The simplex is *regular* if all its edges have the same length, i.e., all the pairwise distances between its vertices are equal. In such a case, we have the following relation between its edge length $a$ and the radius $r$ of its smallest circumscribing sphere:

$$a = r\sqrt{\frac{2(k+1)}{k}}. \tag{7}$$

Theorem 4.1 explains the behavior of T-REG as follows: first, minimizing the term $\mathcal{L}_E$ in Equation (6) expands the point cloud until the sphere constraint term $\mathcal{L}_S$ becomes the dominating term (which happens eventually since $\mathcal{L}_S$ grows quadratically with the scaling factor, versus linearly for $\mathcal{L}_E$); at that stage, the points stop expanding and start spreading themselves out uniformly along the sphere of directions. The amount of expansion before spreading is prescribed by the strength of the sphere constraint term versus the term in the loss, which is driven by the ratio between their respective mixing parameters $\lambda$ and $\gamma$.

The proof of Theorem 4.1 relies on the following two ingredients: a standard result in convex geometry (Proposition 4.2), and a technical lemma—proved in Appendix A—relating the length of the MST to the sum of pairwise distances (Lemma 4.3).

**Proposition 4.2** (Eq. (14.25) in Apostol and Mnatsakanian [3]). *Under the conditions of Theorem 4.1, and assuming $n = d + 1$, the sum of pairwise distances $\sum_{1 \leq i < j \leq n} \|z_i - z_j\|_2$ is maximal when the points of $X$ lie on the bounding sphere $S$, at the vertices of a regular $d$-simplex.*

**Lemma 4.3.** *For any points $z_1, \ldots, z_n \in \mathbb{R}^d$:*

$$E\left(\mathrm{MST}\left(\{z_1, \ldots, z_n\}\right)\right) \leq \frac{2}{n} \sum_{1 \leq i < j \leq n} \|z_i - z_j\|_2.$$

*Proof of Theorem 4.1.* We prove the result in the case $n = d + 1$. The case $n < d + 1$ is the same modulo some extra technicalities and can be found in Appendix A.

Let $z_1^*, \ldots, z_n^* \in \mathbb{S}$ lie at the vertices of a regular $d$-simplex. Then, for any points $z_1, \ldots, z_n \in B$:

$$
\begin{aligned}
E\left(\mathrm{MST}\left(\{z_1, \ldots, z_n\}\right)\right) \quad &\overset{\text{Lemma 4.3}}{\leq} \quad \frac{2}{n} \sum_{1 \leq i < j \leq n} \|z_i - z_j\|_2 \\[2mm]
&\overset{\text{Proposition 4.2}}{\leq} \quad \frac{2}{n} \sum_{1 \leq i < j \leq n} \left\|z_i^* - z_j^*\right\|_2 \\[2mm]
&\overset{\text{Eq. (7)}}{=} \quad \frac{2}{n} \frac{n(n-1)}{2} \, r\sqrt{\frac{2(d+1)}{d}} = (n-1)\, r\sqrt{\frac{2(d+1)}{d}} \\[2mm]
&= \quad E\left(\mathrm{MST}\left(\{z_1^*, \ldots, z_n^*\}\right)\right).
\end{aligned}
$$

$\square$

### 4.1.2 Asymptotic behavior on large samples

We now consider the case where $n > d + 1$, focusing specifically on the asymptotic behavior as $n \to \infty$. We analyze the constant $C$ in Equation (2), which can be made independent of the density of the sampling $X$. This, in particular, allows us to show that uniform and dimension-maximizing densities are asymptotically optimal for $E(\mathrm{MST}(\cdot))$. We fix a compact Riemannian $d$-manifold, $\mathcal{M}$, equipped with the $d$-dimensional Hausdorff measure $\mu$.

**Theorem 4.4** ([15, Corollary 5]). *Let $X_n$ be an iid $n$-sample of a probability measure on $\mathcal{M}$ with density $f_X$ w.r.t. $\mu$. Then, there exists a constant $C'$ independent of $f_X$ and of $\mathcal{M}$ such that:*

$$n^{-\frac{d-1}{d}} \cdot E\left(\mathrm{MST}(X_n)\right) \xrightarrow[n \to \infty]{} C' \int f_X^{\frac{d-1}{d}} \, \mathrm{d}\mu \quad \text{almost surely.} \tag{8}$$

As pointed out by Costa and Hero [15], the limit in Equation (8) is related to the *intrinsic Rényi $\frac{d-1}{d}$-entropy*:

$$\varphi_{\frac{d-1}{d}}(f) = \frac{1}{1 - \frac{d-1}{d}} \log \int f^{\frac{d-1}{d}} \, \mathrm{d}\mu, \tag{9}$$

which is known to converge to the Shannon entropy as $\frac{d-1}{d} \to 1$ [6]. The Shannon entropy, in turn, achieves its maximum at the uniform distribution on compact sets [48]. This result can be shown by directly studying the map $\phi \colon f \mapsto \int f^p \, \mathrm{d}\mu$, which shows that an optimal density function $f_X$ maximizes the dimensionality of the sampling $X_n$. Given a compact set $K \subseteq \mathcal{M}$, we consider the space $\mathcal{D}_K$ of positive, continuous probability densities $f$ on $K$.

**Proposition 4.5.** *For any $0 < p < 1$ and any compact set $K \subseteq \mathcal{M}$, the map $\phi|_{\mathcal{D}_K}$ admits a unique maximum at the uniform distribution $U_K$ on $K$. Furthermore, we have $\phi(U_A) < \phi(U_B)$ for all sets $A, B \subseteq M$ such that $\mu(A) < \mu(B)$.*

*Proof.* The map $\phi$ is strictly concave, as the composition of the strictly concave function $x \mapsto x^p$ with the linear map $x \mapsto \int x \, \mathrm{d}\mu$. Since density functions integrate to 1 over $K$, maximizing $\phi$ corresponds to an optimization problem under constraint, which can be solved using the Lagrangian:

$$\mathcal{L}_\lambda(f) := \int_K f^p \, \mathrm{d}\mu - \lambda \left( \int_K f \, \mathrm{d}\mu - 1 \right), \quad \text{with differential } \mathrm{d}\left(\mathcal{L}_\lambda\right)_f (h) = \int_K \left( p f^{p-1} - \lambda \right) h \, \mathrm{d}\mu.$$

Then, $\mathrm{d}\left(\mathcal{L}_\lambda\right)_f = 0$ is equivalent to $pf^{p-1} - \lambda$ being $0$ almost everywhere, i.e. $f$ being equal to $\left(\frac{\lambda}{p}\right)^{\frac{1}{p-1}}$ (a constant determined by $\int f \,\mathrm{d}\mu = 1$) almost everywhere, i.e. $f$ being the density of the uniform measure on $K$.

Now, recalling that $0 < p < 1$, we have:

$$\phi(U_K) = \int \left(\frac{\mathrm{d}U_K}{\mathrm{d}\mu}\right)^p \mathrm{d}\mu = \int_K \frac{1}{\mu(K)^p} \,\mathrm{d}\mu = \int \frac{\mu(K)}{(\mu(K))^p} \,\mathrm{d}U_K = \mu(K)^{1-p},$$

which proves the second part of the statement. $\qquad\square$

A direct consequence of Proposition 4.5 and the fact that $\mathcal{D}_\mathcal{M}$ is dense in the set of probability densities of $L^p(\mathcal{M})$ is the following corollary.

**Corollary 4.6.** *Let $\mathcal{D}$ be the set of probability densities over $\mathcal{M}$, and $p \in (0, 1)$. Then, the map $f \in \mathcal{D} \mapsto \int f^p \,\mathrm{d}\mu$ reaches its maximum at the density $f := \frac{\mathrm{d}U_\mathcal{M}}{\mathrm{d}\mu}$.*

### 4.2  Empirical evaluation

We conduct an empirical study on synthetic data to validate T-REG's ability to prevent dimensional collapse and promote sample uniformity.

**Preventing dimensional collapse.** Following Fang et al. [23], we assess T-REG's effectiveness against dimensional collapse by measuring how sensitive its loss $\mathcal{L}_E$ is to simulated collapse. Specifically, we generate $10{,}000$ data points in dimension $1024$ from an isotropic Gaussian distribution, then zero out a fraction $\eta$ of their coordinates to control the collapse level. As shown in Figure 3, the sensitivity of the T-REG loss to $\eta$ is similar to that of the $\mathcal{W}_2$ loss from Fang et al. [23], indicating that T-REG effectively penalizes dimensional collapse.

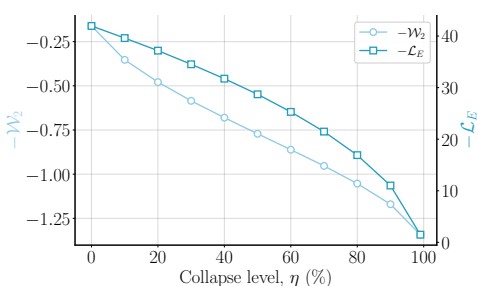

Figure 3: **Sensitivity to Dimensional Collapse.** The metrics $-\mathcal{W}_2$ and $-\mathcal{L}_E$ jointly decrease as the collapse level ($\eta$) increases.

**Promoting sample uniformity.** We apply T-REG alone to optimize a given point cloud and analyze its behavior in both low-dimensional and high-dimensional scenarios (Figure 2). For this, we use two different input point clouds: *(i)* a degenerate set of 256 points on a 1-d curve (corresponding to the orange dots in Figures 2a and 2b); *(ii)* a set of 256 points in $\mathbb{R}^{256}$, initially concentrated around a specific point on the unit sphere, where each point $x_i$ is sampled as $x_i = e_1 + \varepsilon_i$, with $\varepsilon_i$ drawn uniformly from a ball of radius $0.001$.

As illustrated in Figure 2a, optimization with T-REG successfully transforms the initial point cloud in a 3-d space into a uniformly distributed point cloud on the sphere, as per Corollary 4.6. This is achieved through the combination of MST length maximization and sphere constraint. The sphere constraint is crucial here: when optimizing only the MST length, $\mathcal{L}_E$, (see Figure 2b), the optimization fails to converge.

In high dimensions (Figure 2d), we analyze the distribution of cosine similarities between the embeddings. The initial distribution shows a sharp peak near 1, indicating highly correlated samples on the sphere. After optimization with T-REG, the distribution becomes almost a Dirac slightly below 0, indicating that the configuration of the points is close to that of the vertices of the regular simplex, as per Theorem 4.1.

## 5  T-REGS: T-REG for Self-supervised learning

**T-REGS** extends T-REG to Joint-Embedding Self-Supervised Learning. For an input image $i$, two transformations $t, t'$ are sampled from a distribution $\mathcal{T}$ to produce two augmented views $x = t(i)$ and $x' = t'(i)$. These transformations are typically random crops and color distortions. We compute

| Method | | CIFAR-10 [37] | CIFAR-100 [37] |
|---|---|---|---|
| Zero-CL [68] | | 91.3 | **68.5** |
| | $+ \mathcal{L}_u$ | 91.3 | 68.4 |
| | $+ \mathcal{W}_2$ | 91.4 | **68.5** |
| MoCo v2 [14] | | 90.7 | 60.3 |
| | $+ \mathcal{L}_u$ | 91.0 | 61.2 |
| | $+ \mathcal{W}_2$ | 91.4 | 63.7 |
| BYOL [28] | | 89.5 | 63.7 |
| | $+ \mathcal{L}_u$ | 90.1 | 62.7 |
| | $+ \mathcal{W}_2$ | 90.1 | 65.2 |
| | $+ \mathcal{L}_{\text{T-REGS}}$ | 90.4 | 65.7 |
| Barlow Twins [65] | | 91.2 | 68.2 |
| | $+ \mathcal{L}_u$ | 91.4 | 68.4 |
| | $+ \mathcal{W}_2$ | 91.4 | **68.5** |
| | $+ \mathcal{L}_{\text{T-REGS}}$ | **91.8** | **68.5** |
| $\mathcal{L}_{\text{MSE}}$ | $+ \mathcal{L}_{\text{T-REGS}}$ | 91.3 | 67.4 |

Table 1: **Comparison with $\mathcal{W}_2$ regularization [23] on CIFAR-10/100.** The table is inherited from Fang et al. [23], and we follow the same protocol: ResNet-18 models are pre-trained for 500 epochs on CIFAR-10/100, with a batch size of 256, followed by linear probing. We report Top-1 accuracy (%). **Boldface** indicates best performance.

embeddings $z = h_\phi(f_\theta(x))$ and $z' = h_\phi(f_\theta(x'))$ using a backbone $f_\theta$ and projector $h_\phi$. T-REGS acts as a regularization applied separately to the embedding batches $Z = [z_1, ..., z_n]$ and $Z' = [z'_1, ..., z'_n]$. Specifically, embeddings from each view batch, $Z$ and $Z'$, are treated as points in a high-dimensional space, and Kruskal's algorithm [38] is used to construct two Minimum Spanning Tree (MST), one for each view batch. These MSTs yield two T-REG regularization terms, which are combined into the T-REGS objective as follows:

$$\mathcal{L}_{\text{T-REGS}}(Z, Z') = \underbrace{\gamma \, \mathcal{L}_{\text{E}}(Z) + \lambda \, \mathcal{L}_{\text{S}}(Z)}_{\mathcal{L}_{\text{T-REG}}(Z)} + \underbrace{\gamma \, \mathcal{L}_{\text{E}}(Z') + \lambda \, \mathcal{L}_{\text{S}}(Z')}_{\mathcal{L}_{\text{T-REG}}(Z')}. \qquad (10)$$

where $\gamma, \lambda$ control the contribution of each term.

In practice, T-REGS can be used as *(i)* a standalone regularization, combined directly with an invariance term such as the Mean Squared Error: $\mathcal{L}(Z, Z') = \beta \, \mathcal{L}_{\text{MSE}}(Z, Z') + \mathcal{L}_{\text{T-REGS}}(Z, Z')$, where $\mathcal{L}_{\text{MSE}}(Z, Z') = \frac{1}{n} \sum_i \|z_i - z'_i\|_2^2$ and $\beta$ is a mixing parameter; or *(ii)* as an auxiliary loss to existing SSL methods: $\mathcal{L}(Z, Z') = \beta \, \mathcal{L}_{\text{SSL}}(Z, Z') + \mathcal{L}_{\text{T-REGS}}(Z, Z')$, where $\mathcal{L}_{\text{SSL}}(Z, Z')$ denotes the objective function of a given SSL method, and $\beta$ is a mixing parameter. An overview of T-REGS is presented in Figure 1.

The remainder of the section provides evaluations of T-REGS on standard SSL benchmarks (Section 5.1) and on a multi-modal application (Section 5.2); as well as loss coefficients and computational analyses (Section 5.3). Implementation details and further analyses are in Appendices C and E.

## 5.1 Evaluation on standard SSL benchmark

We evaluate the representations obtained after training with T-REGS, either directly combined with view invariance or integrated with existing methods (i.e., BYOL, and Barlow Twins) on CIFAR-10/100 [37], ImageNet-100 [59], and ImageNet [18]. Our implementation is based on `solo-learn` [16], and we use `torchph` [7] for the MST computations. For T-REGS as a standalone regularizer, we use $\beta = 10$, $\gamma = 0.2$, $\lambda = 8e - 4$.

**Evaluation on CIFAR-10/100.** We first focus on comparisons with Fang et al. [23] ($\mathcal{W}_2$-regularized methods), following the same protocol on CIFAR-10/100 [37] with ResNet-18. As shown in Table 1, T-REGS demonstrates strong standalone performance, achieving results within $0.1\%$ of the best $\mathcal{W}_2$-regularized approach on CIFAR-10. Additionally, using T-REGS as an auxiliary loss consistently improves performance over the respective baselines, and over variants that use $\mathcal{L}_u$ or $\mathcal{W}_2$ as additional regularization terms.

**Evaluation on ImageNet-100/1k.** To assess the scalability of T-REGS, we evaluate our model on ImageNet-100 and ImageNet-1k using ResNet-18 and ResNet-50, respectively, following the standard linear evaluation protocol on ImageNet and comparing with the state of the art. We report Top-1 accuracy. As shown in Table 2, T-REGS is competitive with methods that use the same number of views (e.g., INTL), and improves existing methods when used as an auxiliary loss.

| # views | Method | | Imagenet-100 [59] Top-1 | ImageNet-1k [18] Batch Size | Top-1 |
|---|---|---|---|---|---|
| 8 | SwAV [9] | | 74.3 | 4096 | 66.5 |
| | FroSSL [56] | | 79.8 | - | - |
| | SSOLE [34] | | **82.5** | 256 | **73.9** |
| 2 | SimCLR [12] | | 77.0 | 4096 | 66.5 |
| | MoCo v2 [14] | | 79.3 | 256 | 67.4 |
| | SimSiam [13] | | 78.7 | 256 | 68.1 |
| | W-MSE [20] | | 69.1 | 512 | 65.1 |
| | Zero-CL [68] | | 79.3 | 1024 | **68.9** |
| | VICReg [4] | | 79.4 | 1024 | 68.3 |
| | CW-RGP [62] | | 77.0 | 512 | 67.1 |
| | INTL [63] | | **81.7** | 512 | **69.5** |
| | BYOL [28] | | 80.3 | 1024 | 66.5 |
| | | + $\mathcal{L}_{\text{T-REGS}}$ | **80.8** | 1024 | 67.2 |
| | Barlow Twins [65] | | 80.2 | 2048 | 67.7 |
| | | + $\mathcal{L}_{\text{T-REGS}}$ | **80.9** | 2048 | 67.8 |
| | $\mathcal{L}_{\text{MSE}}$ | + $\mathcal{L}_{\text{T-REGS}}$ | 80.3 | 512 | **68.8** |

Table 2: **Linear Evaluation on ImageNet-100/1k.** We report Top-1 accuracy (%). The top-4 methods are boldfaced. For ImageNet-100, ResNet-18 are pre-trained for 400 epochs using a batch size of 256; for ImageNet-1k, ResNet-50 is pre-trained for 100 epochs. The table is mostly inherited from Weng et al. [63].

## 5.2 Evaluation on Multi-modal application: image-text retrieval

| | Flickr30k [49] | | | | MS-COCO [43] | | | |
|---|---|---|---|---|---|---|---|---|
| | i → t | | t → i | | t → i | | t → i | |
| Method | R@1 | R@5 | R@1 | R@5 | R@1 | R@5 | R@1 | R@5 |
| Zero-Shot | 71.1 | 90.4 | 68.5 | 88.9 | 31.9 | 56.9 | 28.5 | 53.1 |
| Finetune | 81.2 | 95.5 | 80.7 | 95.8 | 36.7 | 63.6 | 36.9 | 63.9 |
| ES [41] | 71.8 | 90.0 | 68.5 | 88.9 | 31.9 | 56.9 | 28.7 | 53.0 |
| $i$-Mix [39] | 72.3 | 91.7 | 69.0 | 91.1 | 34.0 | 63.0 | 34.6 | 62.2 |
| Un-Mix [54] | 78.5 | 95.4 | 74.1 | 91.8 | 38.8 | 66.2 | 33.4 | 61.0 |
| $m^3$-Mix [45] | 82.3 | 95.9 | **82.7** | 96.0 | 41.0 | 68.3 | 39.9 | 67.9 |
| $\mathcal{L}_{\text{CLIP}} + \mathcal{L}_{\text{T-REGS}}$ | **83.2** | **96.0** | 80.8 | **96.4** | **41.6** | **68.7** | **41.5** | **68.7** |

Table 3: **Cross-Modal Retrieval after finetuning CLIP.** Image-to-text (i → t) and text-to-image (t → i) retrieval results (top 1/5 Recall: R@1, R@5). The table is mostly inherited from Oh et al. [45]. Boldface indicates the best performance.

T-REGS can also be applied when branches differ in architecture and data modalities, as it regularizes each branch independently. Accordingly, we demonstrate its capabilities in a joint-embedding multi-modal setting.

Pre-trained multi-modal models, such as CLIP [51], provide broadly transferable embeddings. However, several works have shown that CLIP preserves distinct subspaces for text and image—the *modality gap* [42, 45, 36]. Prior analyses [45, 64] relate this gap to low embedding uniformity; notably, CLIP's embedding space often remains non-uniform even after fine-tuning, which can hinder transferability. Given that T-REGS improves embedding uniformity when used as an auxiliary loss, we evaluate its impact on CLIP fine-tuning. We fine-tune CLIP using T-REGS as an auxiliary regularizer; more precisely, $\mathcal{L}_{\text{T-REGS}}$ is applied independently to the image and text branches and combined with the standard $\mathcal{L}_{\text{CLIP}}$ objective [51] to encourage more robust and uniformly distributed representations. We follow the protocol of $m^3$-Mix [45]. We report R@1 and R@5 for image-to-text and text-to-image retrieval on Flickr30k and MS-COCO in Table 3, which shows that T-REGS improves performance over prior methods.

## 5.3 Analysis

**Loss coefficients.** We determine the final coefficients for T-REGS as a standalone regularizer on ImageNet-1k as follows (the same approach was applied when combining T-REGS with existing methods). Initial experiments revealed that maintaining $\beta \geq \gamma \geq \lambda$ was essential to prevent representation collapse. To efficiently explore the parameter space while managing computational

| Coefficients | | | Scaling | | |
|---|---|---|---|---|---|
| $\beta$ | $\gamma$ | $\lambda$ | $\frac{\beta}{\gamma}$ | $\frac{\gamma}{\lambda}$ | **Top-1** |
| 1 | - | - | - | - | collapse |
| 1 | 1 | - | 1 | - | collapse |
| 10 | 1 | 1 | 10 | 1 | collapse |
| 10 | 0.5 | 5e-2 | 20 | 10 | 25.7 |
| 10 | 0.2 | 2e-2 | 50 | 10 | 45.4 |
| 10 | 0.5 | 2.5e-3 | 20 | 200 | 65.0 |
| 10 | 0.2 | 1e-3 | 50 | 200 | 65.3 |
| 10 | 0.5 | 2e-3 | 20 | 250 | 64.9 |
| 10 | 0.2 | 8e-4 | 50 | 250 | **66.1** |
| 10 | 0.02 | 8e-5 | 100 | 300 | 63.3 |

Table 4: **Impact of coefficients.** $\mathcal{L}_{\text{MSE}}$ +$\mathcal{L}_{\text{T-REGS}}$ top-1 accuracy (%) on ImageNet-1k with online evaluation protocol over 50 epochs. **Boldface** indicates best performance.

| Method | Complexity | $B$ range | $D$ range | Wall-clock time |
|---|---|---|---|---|
| SimCLR [12] | $\mathcal{O}(B^2 \cdot D)$ | [2048-4096] | [256-1024] | $0.22 \pm 0.03$ |
| VICReg [4] | $\mathcal{O}(B \cdot D^2)$ | [1024-4096] | [4096-8192] | $0.23 \pm 0.02$ |
| $\mathcal{L}_{\text{MSE}} + \mathcal{L}_{\text{T-REGS}}$ | $\mathcal{O}(B^2(D \cdot \log B))$ | [512-1024] | [512-2048] | $0.20 \pm 0.001$ |

Table 5: **Complexity and computational cost.** Comparison between different methods is performed, with training on ImageNet-1k distributed across 4 Tesla H100 GPUs. The wall-clock time (sec/step) is averaged over 500 steps. $B, D$ ranges are reported from Bardes et al. [4], Garrido et al. [25].

costs, we fixed $\beta$ (the largest coefficient) and systematically varied the ratios $\frac{\beta}{\gamma}$ and $\frac{\gamma}{\lambda}$ using 50-epoch online probing [25]. As shown in Table 4, both $\mathcal{L}_{\text{E}}$ and $\mathcal{L}_{\text{S}}$ contribute to performance, with $\mathcal{L}_{\text{S}}$ requiring a smaller weight. This suggests that the actual radius of the sphere is not critical, thereby validating our choice of a soft sphere constraint instead of a hard one.

**Computational cost.** We evaluate the computational cost of T-REGS. The MSTs are computed with Kruskal's algorithm [38], whose worst-case time is $\mathcal{O}(B^2(D \cdot \log B))$, with $B$ the batch size and $D$ the embedding dimension. Although Kruskal's main loop is sequential, preprocessing (computing the distance matrix and sorting its entries) dominates in practice and can be efficiently parallelized on GPUs (as in `torchph` [7], used in our implementation). Empirically, T-REGS matches the per-step wall-clock of VICReg and SimCLR (averaged over 500 steps during training on ImageNet-1k with $B = 512, D = 1024$; see Table 5).

## 6 Conclusion

We introduced T-REG, a regularization approach that prevents dimensional collapse and promotes sample uniformity. Our method maximizes the length of the minimum spanning tree (MST), coupled with a sphere constraint. Our analysis connects MST optimization to entropy maximization and uniformity on compact manifolds, providing theoretical guarantees corroborated by empirical results. We extend T-REG to Self-supervised learning, yielding T-REGS. On CIFAR-10/100 and ImageNet-100/1k, T-REGS is competitive with $\mathcal{W}_2$-regularized and state-of-the-art methods, both as a standalone regularizer and as an auxiliary term, underscoring its effectiveness.

## Acknowledgments

This work was partially supported by Inria Action Exploratoire PREMEDIT (Precision Medicine using Topology), a Hi!Paris grant and ANR/France 2030 program. We were granted access to the HPC resources of IDRIS under the allocations 2024-AD011014747R1 and 2025-AD011016121 made by GENCI. We would like to thank Robin Courant for his helpful feedback.

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

**Appendix to**

# T-REGS: Minimum Spanning Tree Regularization for Self-Supervised Learning

## Table of Contents

## A  Missing proofs from Section 4.1

*Proof of Theorem 4.1 (case $n < d + 1$).* Let $z_1, \ldots, z_n \in B$, and let $H \subset \mathbb{R}^d$ be an $n$-dimensional affine space containing $z_1, \ldots, z_n$. Let $B_H$ be the $(n-1)$-dimensional Euclidean ball $B \cap H$, and $S_H = S \cap H$ its bounding sphere. Finally, let $r_H \leq r$ be the radius of $B_H$ (and of $S_H$). Inside $H$, the same calculation as in the case $n = d + 1$ shows that

$$E\left(\text{MST}\left(\{z_1, \ldots, z_n\}\right)\right) \leq E\left(\text{MST}\left(\{z_1^*, \ldots, z_n^*\}\right)\right) = (n-1)\, r_H \sqrt{\frac{2n}{n-1}}$$

for any points $z_1^*, \ldots, z_n^*$ lying at the vertices of a regular $(n-1)$-simplex inscribed in $S_H$. The quantity on the right-hand side of the equality is bounded above by $(n-1)r\sqrt{\frac{2n}{n-1}}$, which is attained when $r_H = r$, i.e., when the affine subspace $H$ contains the origin, or equivalently, when $S$ is the smallest circumscribing sphere of $z_1^*, \ldots, z_n^*$. $\qquad\square$

*Proof of Lemma 4.3.* We fix $d$ and proceed by induction on $n$.

For $n = 1$ we have:

$$E\left(\text{MST}\left(\{z_1\}\right)\right) = 0 = \frac{2}{1} \sum_{1 \leq i < j \leq n} \|z_i - z_j\|_2 \,.$$

Assume now that the result holds for all $n$ up to some $n_0 \geq 1$, and let us prove it for $n = n_0 + 1$. Let $e = (z_i, z_j)$ be an edge of MST $(\{z_1, \ldots, z_n\})$ of maximum length. Then, the graph $G = $ MST $(\{z_1, \ldots, z_n\}) \setminus \{e\}$ has two connected components $C$ and $D$, each of which is a tree with less than $n$ vertices. Up to a relabeling of the points of $Z$, we can assume without loss of generality that $j = i + 1$ and that the vertices of $C$ are the points $z_1, \ldots, z_i$ while the vertices of $D$ are the points $z_{i+1}, \ldots, z_n$. We can then make the following observations:

(i) $C = $ MST $(\{z_1, \ldots, z_i\})$ and $D = $ MST $(\{z_{i+1}, \ldots, z_n\})$. Indeed, otherwise, replacing $C$ by MST $(\{z_1, \ldots, z_i\})$ and $D$ by MST $(\{z_{i+1}, \ldots, z_n\})$, and connecting them with edge $e$, would yield a spanning tree of $\{z_1, \ldots, z_n\}$ of strictly smaller length than $G \cup \{e\}$, which would contradict the fact that $G \cup \{e\} = $ MST $(\{z_1, \ldots, z_n\})$.

(ii) For all $k \leq i < l$, we have $\|z_k - z_l\|_2 \geq \|z_i - z_{i+1}\|_2$, for otherwise the graph $G \cup \{(z_k, z_l)\}$ would be a spanning tree of $\{z_1, \ldots, z_n\}$ of strictly smaller length than $G \cup \{e\}$, again contradicting the fact that $G \cup \{e\} = $ MST $(\{z_1, \ldots, z_n\})$.

Then, (i) and the induction hypothesis imply:

$$\sum_{k < l \leq i} \|z_k - z_l\|_2 \geq \frac{i}{2} \, E(C), \tag{11}$$

$$\sum_{i < k < l} \|z_k - z_l\|_2 \geq \frac{n - i}{2} \, E(D). \tag{12}$$

Meanwhile, (ii) implies:

$$\sum_{k \leq i < l} \|z_k - z_l\|_2 \geq i(n - i) \, \|z_i - z_{i+1}\|_2 . \tag{13}$$

And since $e = (z_i, z_{i+1})$ is an edge of MST $(\{z_1, \ldots, z_n\})$ of maximum length, we have:

$$\|z_i - z_{i+1}\|_2 \geq \frac{1}{i - 1} \, E(C), \tag{14}$$

$$\|z_i - z_{i+1}\|_2 \geq \frac{1}{n - i - 1} \, E(D). \tag{15}$$

Hence:

$$
\begin{aligned}
\sum_{k \leq i < l} \|z_k - z_l\|_2 & \overset{\text{Eq. (13)}}{\geq} && i(n - i) \, \|z_i - z_{i+1}\|_2 \\
& = && \left( \frac{n}{2} + \frac{(n-i)(i-1)}{2} + \frac{i(n-i-1)}{2} \right) \|z_i - z_{i+1}\|_2 && \text{(16)} \\
& \overset{\text{Eqs. (14)-(15)}}{\geq} && \frac{n}{2} \, \|z_i - z_{i+1}\|_2 + \frac{n-i}{2} \, E(C) + \frac{i}{2} \, E(D).
\end{aligned}
$$

It follows:

$$
\begin{aligned}
\sum_{1 \leq k < l \leq n} \|z_k - z_l\|_2 & = && \sum_{k \leq i < l} \|z_k - z_l\|_2 + \sum_{k < l \leq i} \|z_k - z_l\|_2 + \sum_{i < k < l} \|z_k - z_l\|_2 \\
& \overset{\text{Eqs. (16) and (11)-(12)}}{\geq} && \frac{n}{2} \, \|z_i - z_{i+1}\|_2 + \frac{n-i}{2} \, E(C) + \frac{i}{2} \, E(D) + \frac{i}{2} \, E(C) + \frac{n-i}{2} \, E(D) \\
& = && \frac{n}{2} \, \|z_i - z_{i+1}\|_2 + \frac{n}{2} \, E(C) + \frac{n}{2} \, E(D) \\
& = && \frac{n}{2} \, E \left( \text{MST} (\{z_1, \ldots, z_n\}) \right).
\end{aligned}
$$

$\square$

# B    Algorithm

---

**Algorithm 1** T-REGS combined with view invariance using PyTorch pseudocode

---

```
# f:  encoder network, h:  projection network
# β, γ, λ:  coefficients of the invariance, MST length and sphere constraint
losses

for x in loader: do # load a batch with N samples
   # two randomly augmented versions of x
   x_1, x_2 = augment(x), augment(x)
   # compute the representations and the embeddings
   y_1, y_2 = f(x_1), f(x_2)
   z_1, z_2 = h(y_1), h(y_2)

   inv_loss = L_MSE(z_1,z_2) # invariance loss
   length_mst_loss = L_E(z_1) + L_E(z_2) # MST length
   sphere_loss = L_S(z_1) + L_S(z_2) # soft sphere constraint loss
   # total loss
   loss = β inv_loss + γ length_mst_loss + λ sphere_loss

   # optimization step
   loss.backward()
   optimizer.step()
end for
```

---

# C    Implementation Details on standard SSL Benchmark

Our implementation is based on `solo-learn` [16], which is released under the MIT License. To compute the length of the minimum spanning tree, we rely on `torchph` and `Gudhi` [50], both released under MIT Licenses.

Our experiments are performed on

1. ImageNet dataset [18], and a subset ImageNet-100 which are subject to the ImageNet terms of access

2. CIFAR-10, CIFAR-100

## C.1    Architectural and training details.

| | **CIFAR-10** [37] | **CIFAR-100** [37] | **Imagenet-100** [59] | **ImageNet** [18] |
|---|---|---|---|---|
| *Backbone* | | | | |
| backbone | Resnet-18 | Resnet-18 | Resnet-18 | Resnet-50 |
| *Projector* | | | | |
| projector layers | | 3 layers with BN and ReLU | | |
| projector hidden dimension | 2048 | 2048 | 4096 | 4096 |
| projector output dimension | | 1024 | | |
| *Pre-training* | | | | |
| batch size | 256 | 256 | 256 | 512 |
| optimizer | | LARS | | |
| learning rate | | base_lr $*$ batch size$/256$ | | |
| base_lr | 0.4 | 0.4 | 0.3 | 1 |
| learning rate warm-up | | 2 epochs | | 10 epochs |
| learning rate schedule | | cosine decay | | |
| weight decay | | 1e-4 | | 1e-5 |
| *Linear evaluation* | | | | |
| batch size | | 256 | | |
| optimizer | | SGD | | |
| base_lr | | 0.1 | | |
| learning rate schedule | | cosine decay | | |

Table 6: **Training hyperparameters.**

We follow the guidance of Da Costa et al. [16] for selecting baseline hyperparameters and use the same seed: 5. Table 6 lists each dataset's architectural and training details.

## C.2 Augmentations

We follow the image augmentation protocol first introduced in SimCLR [12] and now commonly used in Joint-Embedding Self-Supervised Learning [4, 9, 65]. Two random crops from the input image are sampled and resized to $32 \times 32$ for CIFAR-10/100 and $224 \times 224$ for Imagenet-100/1k, followed by color jittering, converting to grayscale, Gaussian blurring, polarization, and horizontal flipping. Each crop is normalized in each color channel using the ImageNet mean and standard deviation pixel values. The following operations are performed sequentially to produce each view:

**ImageNet-1k Data Augmentation.**

- Random cropping with an area uniformly sampled with size ratio between 0.2 to 1.0, followed by resizing to size $224 \times 224$.
- Color jittering of brightness, contrast, saturation and hue, with probability 0.8.
- Grayscale with probability 0.2.
- Gaussian blur with probability 0.5 and kernel size 23.
- Solarization with probability 0.1.
- Random horizontal flip with probability 0.5.
- Color normalization using ImageNet mean and standard deviation pixel values (with mean (0.485, 0.456, 0.406) and standard deviation (0.229, 0.224, 0.225).)

**CIFAR-10/100 Data Augmentation.**

- Random cropping with an area uniformly sampled with size ratio between 0.2 to 1.0, followed by resizing to size $32 \times 32$.
- Color jittering of brightness, contrast, saturation and hue, with probability 0.8.
- Grayscale with probability 0.2.
- Solarization with probability 0.1.
- Random horizontal flip with probability 0.5.
- Color normalization with mean (0.485, 0.456, 0.406) and standard deviation (0.229, 0.224, 0.225).

## C.3 Implementation details on Image-text retrieval

Following Oh et al. [45], we fine-tune CLIP ViT-B/32 on Flickr30k and MS COCO, respectively. We train for 9 epochs with a batch size of 128 using the Adam optimizer ($\beta_1 = 0.9, \beta_2 = 0.98, \varepsilon = 1e-6$). We search for the best initial learning rate from $\{1e-6, 3e-6, 5e-6, 7e-6, 1e-5\}$ and weight decay from $\{1e-2, 2e-2, 5e-2, 1e-1, 2e-1\}$. For T-REGS combined with $\mathcal{L}_{\text{CLIP}}$, the overall objective is: $\mathcal{L}(Z, Z') = \beta \, \mathcal{L}_{\text{CLIP}}(Z, Z') + \mathcal{L}_{\text{T-REGS}}(Z, Z')$, with $\beta = 1, \gamma = 3e-3, \lambda = 2e-5$.

# D  Compute cost

We conducted our experiments using NVIDIA H100 and V100 GPUs. Training a single T-REGS model requires:

- for ImageNet-1k: 15 hours using 4 H100 GPUs (i.e., amounting to 60 GPU-hours per model);
- for ImageNet-100: 7 hours using 1 H100 GPU;
- for CIFAR-10/100: 7 hours using 1 V100 GPU.

The computational cost for the entire project, including all baseline computations, experiments, hyperparameter tuning, and ablation studies, amounted to approximately 10,000 H100 GPU-hours and 5,000 V100 GPU-hours.

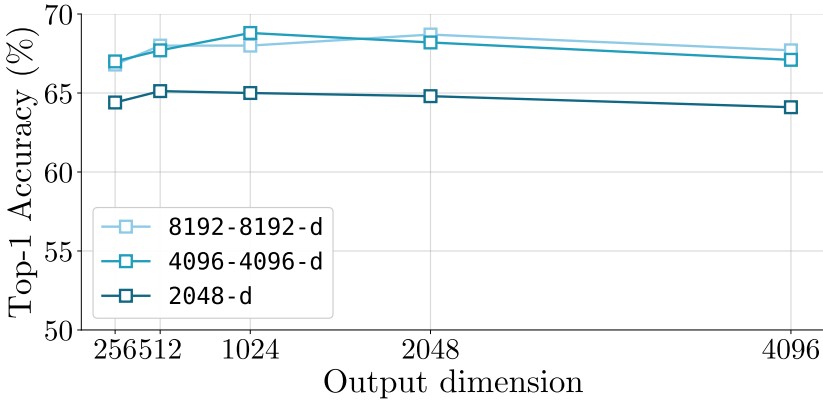

Figure 4: **Impact of the projector architecture.** $\mathcal{L}_{\text{MSE}} + \mathcal{L}_{\text{T-REGS}}$ top-1 accuracy (%) on the linear evaluation protocol with 100 pretraining epochs.

# E    Ablation Study

In this section, we conduct a comprehensive set of ablation experiments to assess the robustness and versatility of T-REGS. These experiments cover various aspects, including projector architecture, batch sizes, and seed variability.

## E.1    Projector Architecture.

An essential difference between methods lies in how the projector ($h_\phi$, in Figure 1) is designed [26]. To assess the impact of the projector architecture on T-REGS performance (when combined with view invariance), we train models for 100 epochs on ImageNet-1k with different projector architectures. We describe projector architectures using the notation `X-Y-Z`, where each number represents the dimension of a linear layer in sequence. Each layer (except the last) is followed by a `ReLU` activation and batch normalization. The final layer has no activation, batch normalization, or bias. We evaluate three projector architectures:

- `2048-d`: the projector used in SimCLR
- `8192-8192-d`: the projector used in VICReg
- `4096-4096-d`: a smaller variant of the VICReg projector

with `d` varying from 256 to 4096. The remaining hyperparameters are the same as reported in Appendix C. Our results demonstrate that the choice of projector architecture significantly impacts the model's performance, with `8192-8192-d` and `4096-4096-d` consistently outperforming the `2048-d` variant.

We also observe that the embedding dimension `d` (i.e., the output dimension of the projector $h_\phi$) has minimal impact on performance. This is particularly noteworthy as other methods, such as VICReg [4] and Barlow Twins [65], are known to be sensitive to embedding dimension, typically requiring dimensions larger than 2048 for optimal performance. Our results show that T-REGS maintains high accuracy even with small embedding dimensions, demonstrating its robustness to this hyperparameter.

## E.2    Batch size.

| Batch Size | 128 | 256 | 512 | 1024 |
|---|---|---|---|---|
| Top-1 | 66.3 | 67.2 | 68.7 | 68.0 |

Table 7: **Impact of batch size.** $\mathcal{L}_{\text{MSE}} + \mathcal{L}_{\text{T-REGS}}$ top-1 accuracy (%) using linear evaluation after 100 pre-training epochs.

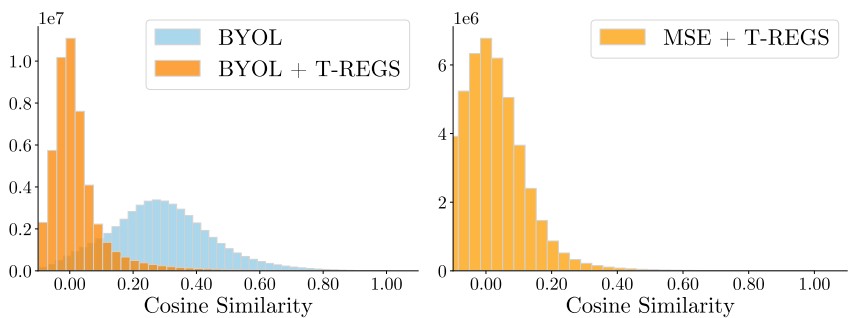

Figure 5: **Histograms of embeddings' cosine similarities on CIFAR-10.** With T-REGS as a standalone regularization (orange) or as an auxiliary loss (dark orange), the distribution of pairwise cosine similarities becomes concentrated around zero, indicating that the embeddings are highly decorrelated and approach a regular simplex configuration (Theorem 4.1).

Many SSL methods are known to be sensitive to batch sizes, for instance contrastive methods suffer from the need of many negative samples which can translate into the need of large batch sizes [12, 9]. In Table 7, we study how batch size affects T-REGS performance (when combined with view invaraince). We train models for 100 epochs on ImageNet-1k with batch sizes ranging from 128 to 1024, using the same hyperparameters as in Table 6. T-REGS maintains good performance even with small batch sizes (e.g., 128), demonstrating its robustness to different batch size configurations.

## E.3 Normalization

Some popular SSL frameworks, such as SimCLR and BYOL, employ explicit normalization of features to the unit sphere, enforcing a hard constraint. Others do not implement such constraints explicitly, and instead rely on soft mechanisms—such as VICReg, which incorporates variance and covariance regularization terms in its loss, implicitly constraining the distribution (and to some extent, the norms) of the embeddings. We chose a soft sphere constraint for the following reasons: *(i)* a hard normalization has been shown to ignore the importance of the embedding norm for gradient computation, whereas a soft constraint enables better embedding optimization [66, 67], *(ii)* during our initial experiment, we found that relaxing the sphere constraint from a hard one to a soft one provides more leeway for optimization and leads to improved results, as shown in Table 8.

|  | CIFAR-10 | CIFAR-100 |
|---|---|---|
| soft-constraint | 91.2 | 66.8 |
| hard-constraint | 89.2 | 64.7 |

Table 8: **Impact of normalization.** $\mathcal{L}_{\text{MSE}} + \mathcal{L}_{\text{T-REGS}}$ top-1 accuracy (%).

## E.4 Sensitivity to the seed.

| CIFAR-10 | CIFAR-100 |
|---|---|
| $91.1 \pm 0.11$ | $66.4 \pm 0.45$ |

Table 9: **Sensitivity to the seed.** $\mathcal{L}_{\text{MSE}} + \mathcal{L}_{\text{T-REGS}}$ top-1 accuracy using linear evaluation after 500 pre-training epochs on CIFAR-10/100. We report results averaged across 5 seeds in the format: `mean ± std`.

We observe strong stability across different random seeds, with standard deviations of only 0.11% and 0.45% on CIFAR-10 and CIFAR-100 respectively.

### E.5 Study of the embeddings.

We analyze the learned embeddings by computing pairwise cosine similarity between embeddings on CIFAR-10. As shown in Figure 5, BYOL yield embeddings with mean similarities significantly above zero ($\approx 0.3$). This indicates a concentration within a cone rather than uniformity on the hypersphere. In contrast, T-REGS yields mean similarities near zero, reflecting more uniformly distributed and decorrelated embeddings, with some values slightly negative– suggesting an arrangement close to a regular simplex, as per Theorem 4.1. Additionally, applying T-REGS as an auxiliary loss effectively shifts the mean cosine similarity towards zero, as illustrated in Figure 5, thus indicating its effectiveness.

## F Empirical Experiments

### F.1 Study of redundancy-reduction methods

**Redundancy-Reduction methods** [4, 65, 63, 20] attempt to produce embedding variables that are decorrelated from each other. These methods maximize the informational content of embeddings by regularizing their empirical covariance matrix.

For instance, VICReg [4] leverages *(i)* a term to encourage the variance (diagonal of the covariance matrix) inside the current batch to be equal to 1, preventing collapse with all the inputs mapped on the same vector; *(ii)* and a correlation regularization, encouraging the off-diagonal coefficients of the empirical covariance matrix to be close to 0, decorrelating the different dimensions of the embeddings. More formally, let $Z = \{z_1, ..., z_n\} \subseteq \mathbb{R}^d$ be a set of $n$ embeddings in $d$-dimensional space. For each dimension $j \in \{1, ..., d\}$, we denote $z^j$ as the vector containing all values at dimension $j$ across the embeddings. The variance term is defined as:

$$\mathcal{L}_{\text{var}} = \frac{1}{d} \sum_{j=1}^{d} \max(0, 1 - S(z^j, \varepsilon)) \tag{17}$$

where $S(x, \varepsilon) = \sqrt{\text{Var}(x) + \varepsilon}$ is a stability-adjusted standard deviation, with $\varepsilon > 0$ being a small constant that prevents numerical instabilities.

The covariance term is defined as:

$$\mathcal{L}_{\text{cov}} = \frac{1}{d} \sum_{i \neq j} [C(Z)]_{i,j}^2 \tag{18}$$

where $C(Z)$ is the sample covariance matrix: $C(Z) = \frac{1}{n-1}(Z - \overline{z})^T (Z - \overline{z})$, with $\overline{z} = \frac{1}{n} \sum_{i=1}^{n} z_i$ the mean value of the embeddings.

The overall variance-covariance regularization term is a weighted:

$$\mathcal{L}_{\text{var-cov}} = \nu \, \mathcal{L}_{\text{var}} + \tau \, \mathcal{L}_{\text{cov}} \tag{19}$$

where $\nu, \tau$ are hyperparameters controlling the importance of each term in the loss.

**Limitations of redundancy-reduction methods.** In Figure 6 we study a limitation of redundancy-reduction methods. We sample 2000 points from a non-isotropic Gaussian distribution, and observe the resulting point cloud after optimization with $\mathcal{L}_{\text{var-cov}}$ (using $\nu = 25, \tau = 1$, as in Bardes et al. [4]). Since Gaussian distributions are fully characterized by their mean and covariance matrix, we expect that by optimizing the empirical covariance matrix, the initial point cloud will converge towards a sample of the standard Gaussian distribution, thus far from a uniform distribution.

Figure 6b shows the optimization of Figure 6a using the covariance-based approach from VICReg (Equation (19)). The optimized empirical distribution is, as expected, closer to the standard Gaussian distribution, but exhibits artifacts such as low-dimensional concentration points or even holes suggesting local instabilities. These artifacts persist across different parameter choices, though their specific manifestations may vary with the learning rate.

Figure 6c shows the optimization of Figure 6a using $\mathcal{L}_{\text{T-REG}}$. We leverage Corollary 4.6, whose guarantees are valid for any Riemannian manifold, and in particular the disk. This ensures that the limiting distribution for T-REG is the uniform distribution on the disk, which prevents dimensional collapse and naturally decorrelates the distribution while maintaining the uniform distribution.

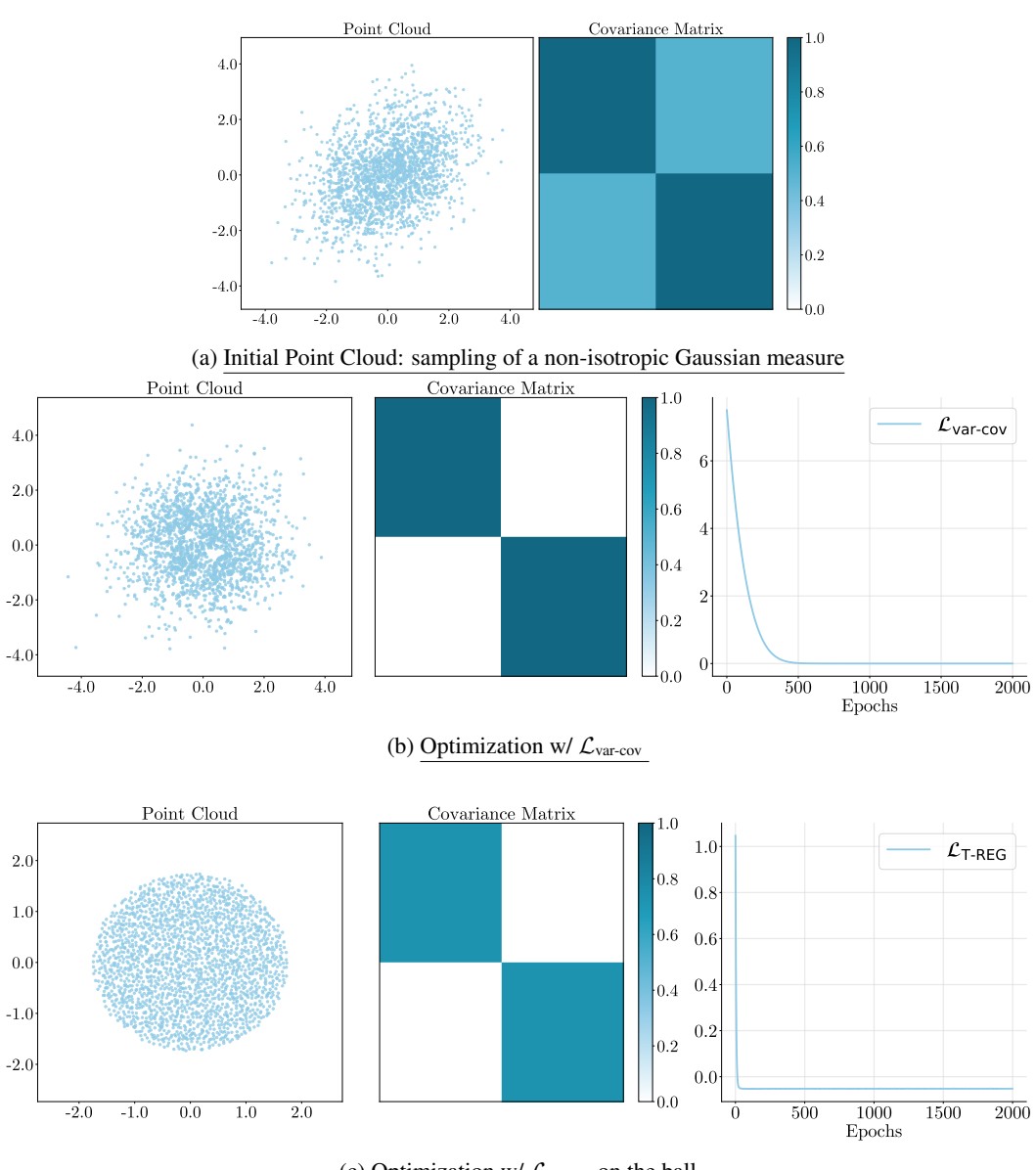

(a) Initial Point Cloud: sampling of a non-isotropic Gaussian measure

(b) Optimization w/ $\mathcal{L}_{\text{var-cov}}$

(c) Optimization w/ $\mathcal{L}_{\text{T-REG}}$ on the ball

Figure 6: **Limitations of redundancy-based methods for non-isotropic Gaussian measure.** (a) Initial sampling from a non-isotropic Gaussian distribution. (b) After $\mathcal{L}_{\text{var-cov}}$ optimization: despite achieving a near-identity covariance matrix (center), the point cloud remains concentrated around its mean, with visible artifacts: holes (left). (c) $\mathcal{L}_{\text{T-REG}}$ optimization achieves both uniform distribution on the disk and near-identity correlation matrix.

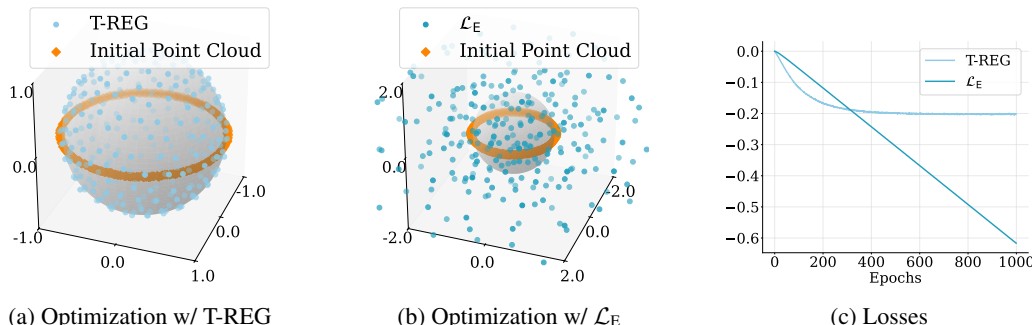

| (a) Optimization w/ T-REG | (b) Optimization w/ $\mathcal{L}_{\mathrm{E}}$ | (c) Losses |

Figure 7: **Further Studying T-REG properties through** $3$**-d point cloud optimization.** (a) T-REG successfully spreads points uniformly on the sphere by combining MST length maximization and sphere constraint, (b) using only MST length maximization leads to excessive dilation, (c) stable convergence of T-REG whereas $\mathcal{L}_{\mathrm{E}}$ does not converge.

## F.2 Promoting sample uniformity

Building upon the empirical study presented in Section 4.2, we conduct an additional point cloud optimization experiment with a different initial configuration (Figure 7). We sample 256 points along a circle while setting the remaining dimension to zero. This setup allows for the generation of a point cloud with one collapsed dimension. The experimental results demonstrate consistent behavior with the findings discussed in Section 4.2: *(i)* T-REG successfully transforms the initial circle into a uniformly distributed point cloud on the sphere (see Figure 7a), *(ii)* the sphere constraint is essential here, since using only the MST length loss (as in Figure 7b) leads to a failure of convergence of the optimization.

This further validates the effectiveness of our approach in promoting uniform point distribution.

## G  Uniformity properties

Recall from Equation (4) that $\mathcal{L}_{\mathrm{E}}(Z) = -E\left(\mathrm{MST}(Z)\right)/|Z|$. In practice, for datasets of fixed size, minimizing $\mathcal{L}_{\mathrm{E}}$ is equivalent to minimizing $-E\left(\mathrm{MST}(Z)\right)$ itself. In this section, we show that, up to a renormalization, $-E\left(\mathrm{MST}(Z)\right)$ satisfies the four principled properties for uniformity metrics introduced in [23, Section 3.1].

A *uniformity metric* $\mathcal{U}$ is a scalar score function on $n$-samples in $\mathbb{R}^d$ that is *large* on uniform-like point clouds and *small* on degenerate or almost-degenerate point clouds. In the following, we fix an $n$-sample $Z = (z_1, \ldots, z_n)$ in $\mathbb{R}^d$.

We define our uniformity metric as $-E\left(\mathrm{MST}(\cdot)\right)$ normalized by the edge length of the regular $d$-simplex $\sigma_d^0$ with vertices on the unit sphere $\mathbb{S}^{d-1}$:

$$\mathcal{U}_{\mathrm{T\text{-}REG}}(Z) := -\frac{E\left(\mathrm{MST}(Z)\right)}{\left(\frac{2(d+1)}{d}\right)^{1/2}}. \tag{20}$$

In practice, for a fixed ambient dimension $d$, minimizing $-E\left(\mathrm{MST}(Z)\right)$ or minimizing $\mathcal{U}_{\mathrm{T\text{-}REG}}(Z)$ is equivalent. The motivation behind renormalization is to measure the length of the minimum spanning tree relative to a *reference* length on the unit sphere $\mathbb{S}^{d-1}$, to make it a dimensionless quantity.

We now show that our uniformity score $\mathcal{U}_{\mathrm{T\text{-}REG}}$ satisfies the desired uniformity properties:

1. *Instance permutation constraint:* $\forall \pi \in \mathfrak{S}_n, \mathcal{U}\left((z_{\pi_1}, \ldots, z_{\pi_n})\right) = \mathcal{U}(Z)$.
   By construction, $\mathrm{MST}(Z)$ and its length are invariant under permutations of the points' indices, therefore $E\left(\mathrm{MST}(\cdot)\right)$ and $\mathcal{U}_{\mathrm{T\text{-}REG}}$ are also permutation invariant.

2. *Instance cloning constraint:* if $Z' = Z$, then $\mathcal{U}\left((z_1, \ldots, z_n, z_1', \ldots, z_n')\right) = \mathcal{U}(Z)$.
   If $G = (Z, E)$ is an MST of $Z$, then

$$G' = ((z_1, \ldots, z_n, z_1', \ldots, z_n'), E \cup \{(z_i, z_i') : 1 \leq i \leq n\}) \tag{21}$$

is a spanning tree of $(z_1, \ldots, z_n, z'_1, \ldots, z'_n)$, and it is minimal for $E$ since $G$ it-self is minimal on $Z$ and $E(G') = E(G) + n \times 0^1 = E(G)$. Therefore, $\mathcal{U}_{\text{T-REG}}((z_1, \ldots, z_n, z_1, \ldots, z_n)) = \mathcal{U}_{\text{T-REG}}(Z)$ since the ambient dimension is constant.

3. *Feature cloning constraint:* $\mathcal{U}(z_1 \oplus z_1, \ldots, z_n \oplus z_n) < \mathcal{U}(z)$.

   Feature cloning corresponds to pushing the points of $Z$ to the diagonal in $\mathbb{R}^{2d}$, which impacts the pairwise distances by a uniform scaling by a factor of $\sqrt{2}$. In particular, the MST remains the same combinatorially and we have:

   $$-E\left(\text{MST}(Z \oplus Z)\right) = -2^{1/2}E\left(\text{MST}(Z)\right) < -E\left(\text{MST}(Z)\right).$$

   Meanwhile, since $\varphi \colon d \mapsto \left(\frac{2(d+1)}{d}\right)^{-1/2}$ is an increasing function and $\mathcal{U}_{\text{T-REG}} \leq 0$, we have:

   $$\mathcal{U}_{\text{T-REG}}(Z \oplus Z) = -E\left(\text{MST}(Z \oplus Z)\right)\varphi(2d) < -E\left(\text{MST}(Z)\right)\varphi(d) = \mathcal{U}_{\text{T-REG}}(Z).$$

4. *Feature baby constraint:* $\forall k \in \mathbb{N}_+, \mathcal{U}(Z \oplus \mathbf{0}^k) < \mathcal{U}(Z)$.

   Adding constant features does not impact the pairwise distances, hence does not impact the minimum spanning tree and its length We thus have $E\left(\text{MST}(Z \oplus \mathbf{0}^k)\right) = E\left(\text{MST}(Z)\right)$.

   As in the previous case, since $\varphi \colon d \mapsto \left(\frac{2(d+1)}{d}\right)^{-1/2}$ is an increasing function and $\mathcal{U}_{\text{T-REG}} \leq 0$, we have:

   $$\mathcal{U}_{\text{T-REG}}(Z \oplus \mathbf{0}^k) = -E\left(\text{MST}(Z \oplus \mathbf{0}^k)\right)\varphi(d+k) < -E\left(\text{MST}(Z)\right)\varphi(d) = \mathcal{U}_{\text{T-REG}}(Z).$$

## H   Example of a Minimum Spanning Tree

Figure 8 provides an example of the MST of a $2d$ uniformly sampled point cloud.

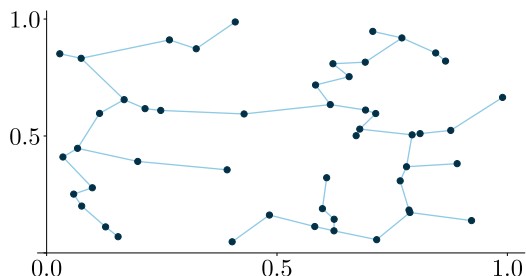

Figure 8: **Example of the** MST **of a 2d uniformly sampled point cloud.**

