# OpenReview forum: "T-REGS: Minimum Spanning Tree Regularization for Self-Supervised Learning"
_NeurIPS.cc/2025/Conference — NeurIPS 2025 spotlight_

### Official Review · Reviewer_hUvh · 2025-06-30

**Clarity:** 3
**Significance:** 3
**Originality:** 3
**Rating:** 4
**Confidence:** 4

**Summary:**

The paper introduces T-REGS, a framework based on Minimum Spanning Tree (MST), to regularize SSL methods to prevent dimensional collapse and promote uniformity. Theoretically, under the assumption of strict sphere constraint, the authors observe that when $n \leq d+1$ ($n$ is the point number and $d$ is the dimension), the optimal MST length is achieved when the embeddings form a simplex. For large samples, the authors first relate the MST length to the  R\'{e}nyi entropy and then propose that uniform distribution maximizes the entropy. In the experiments, T-REG demonstrates the ability the spread the initial point cloud uniformly on the sphere. Additionally, T-REGS yields competitive results across different datasets, though it fails to outperform the baseline.

**Questions:**

1. Are there any results of combining T-REGS with existing SSL methods, both contrastive and non-contrastive? This would further demonstrate the effectiveness of T-REGS.
2. Can T-REGS be generalized to supervised learning?
3. What is the extra cost of MST computation?

**Ethical Concerns:**

["NO or VERY MINOR ethics concerns only"]

**Final Justification:**

During the rebuttal phase, the authors extended the core proposition to a more relaxed assumption (i.e., the points can lie inside the sphere), which makes the proposition more convincing. They also did extra experiments to improve existing results. Besides, they addressed my questions on the results of the experiment on the coefficients of loss functions. Based on these improvements, I have decided to raise my rating to 4.

**Limitations:**

yes

**Quality:**

3

**Strengths And Weaknesses:**

Strengths:
1. The paper provides a detailed theoretical analysis to gain insights of MST and guide the design of T-REG.
2. The authors conduct experiments to demonstrate the effects of T-REG (Figure 2).
3. The idea of this paper is novel and the writing is generally easy to follow.

Weaknesses:
1. The theoretical analysis is based on strict sphere constraints, which cannot be completely ensured in reality. Since regularization is only able to gradually enforce the sphere constraints, a relaxed assumption is more suitable for the setting.
2. There is still a gap between the performance of T-REGS and the baselines, making the effectiveness of the method not convincing enough. More trials regarding the proper choices of the parameters are needed.
3. It is observed in Table 2 that higher values of $\frac{\beta}{\gamma}$ and $\frac{\gamma}{\lambda}$ lead to better performance. Intuitively, this would mean that $\mathcal{L}_E$ and $\mathcal{L}_S$ are not that necessary. To justify the regularization terms, you should first show that no regularization term has poorer performance than your method, and that extremely large $\frac{\beta}{\gamma}$ and $\frac{\gamma}{\lambda}$ also harms the classification accuracy.

---

> ### Author Rebuttal · Authors · 2025-07-29
>
> We thank you for your feedback. Please find below our answers:
>
> **W1. “The theoretical analysis is based on strict sphere constraints, which cannot be completely ensured in reality. Since regularization is only able to gradually enforce the sphere constraints, a relaxed assumption is more suitable for the setting.”**
>
> Thank you for the nice and valuable suggestion! We have extended Proposition 4.1 to the case where the points lie inside the unit ball, not only on the sphere, with the same conclusion: the maximum length of the $\mathrm{MST}$ is achieved by points sitting on the sphere, at the vertices of a regular simplex. This new statement will replace Proposition 4.1 in the final version of the paper. Its proof is actually simpler than the previous one, relying on a standard result in convex geometry—see e.g. Chapter 14, Theorem 14.6, in [1]---to which we add a technical lemma relating the total sum of pairwise distances to the total length of the MST.
>
> The new Proposition 4.1 explains the behavior of T-REG when used with a soft sphere constraint: first, minimizing the term
> $\mathcal L_E$ in Equation (7) expands the point cloud until the sphere constraint $\mathcal L_S$ becomes the dominating term; at that stage, the points stop expanding and start spreading themselves out uniformly along the sphere of directions. The amount of expansion before spreading is prescribed by the strength of the sphere constraint term versus the $\mathrm{MST}$ term in the loss, which is driven by the ratio between their respective mixing parameters $\lambda$ and $\gamma$. Note that $\mathcal L_E$  grows linearly (or sublinearly for $\alpha<1$) with the scale, whereas $\mathcal L_S$ grows quadratically, which explains why $\mathcal L_S$ eventually becomes the dominating term.
>
>
> **W2 & Q1 “There is still a gap between the performance of T-REGS and baselines, making its effectiveness not convincing enough. More trials regarding the proper choices of the parameters are needed.” /  “Are there any results of combining T-REGS with existing SSL, both contrastive and non-contrastive? This would further demonstrate its effectiveness”**
>
> Given the feedback received, we conducted further experiments to combine T-REG with a couple of other regularizers: BYOL and Barlow Twins, on CIFAR-10/100. We provide the results in Table A.2, which highlight the benefits of such combinations (best performances in bold):
>
> #### Table A.2: CIFAR-10/100 Evaluation
> | | CIFAR-10 |CIFAR-100 |
> |-|:-:|:-:|
> | MoCo v2 | 90.7 | 60.3 |
> | MoCo v2 + $\mathcal{L}_u$| 91.0 | 61.2 |
> | MoCo v2 + $\mathcal{W}_2$| 91.4 | 63.7 |
> | ZeroCL | 91.4 | **68.5** |
> | ZeroCL + $\mathcal{L}_u$ | 91.3 | 68.4 |
> | ZeroCL + $\mathcal{W}_2$ | 91.4 | **68.5** |
> | BYOL | 89.5 | 63.7 |
> | BYOL + $\mathcal{L}_u$ | 90.1 |62.7 |
> | BYOL + $\mathcal{W}_2$ | 90.1 |65.2 |
> | BYOL + **T-REG** | 90.4 | 65.7 |
> | Barlow Twins | 91.2 | 68.2 |
> | Barlow Twins + $\mathcal{L}_u$ | 91.4 | 68.4 |
> | Barlow Twins + $\mathcal{W}_2$ | 91.4 | **68.5** |
> | Barlow Twins + **T-REG** | **91.8** | **68.5** |
>
> Adding our regularizer improves the results both for BYOL and Barlow Twins, highlighting its superiority.
>
> **W3. “Table 2: higher values of beta/gamma and gamma/lambda lead to better performance. Intuitively, this would mean that L_E and  L_S are not that necessary. To justify the regularization terms, you should first show that no term has poorer performance than your method, and that extremely large and also harms the accuracy.”**
>
> Following your recommendation, we propose to complete Table 2 as shown below:
>
> #### Table B.1: Impact of coefficients. We report Top-1 accuracy on ImageNet-1k.
> | β  | γ | λ | β/γ | γ/λ | ACC-1 |
> | - |:-:| - | - | - | - |
> | 1 | 0 | 0 | 0 | 0 | collapse |
> | 1 | 1 | 0 | 0 | 0 | collapse |
> | 10 | 1 | 1 | 10 | 1 | collapse |
> | 10 | 0.5 | 5e-2 | 20 | 10 | 25.7 |
> | 10 | 0.2 | 2e-2 | 50 | 10 | 45.4 |
> | 10 | 0.2 | 2.5e-3 | 20 | 200 | 65.0 |
> | 10 | 0.2 | 1e-3 | 50 | 200 | 65.2 |
> | 10 | 0.4 | 2e-3 | 20 | 250 | 64.9 |
> | 10 | 0.2 | 8e-4 | 50 | 250 | 66.1 |
> | 10 | 0.02 | 4e-5 | 500 | 500 | 51.8 |
>
> The new version of the table (Table B.1) more clearly highlights that no regularization at all leads to collapse  (line 1), that large values of regularization also harm classification accuracy (lines 9-10: 51.8% vs best performance of 66.1%), and thus that both $\mathcal L_E$ and $\mathcal L_S$ are essential for optimal performance.
>
>
> **Q2. “Can T-REGS be generalized to supervised learning?”**
>
> The regularization, T-REG, can indeed be applied to other settings beyond the one considered in the paper, when uniformity properties are helpful and/or where dimensional collapse can be an issue. While we do not have an example in supervised learning per se, we have other examples of applications in unsupervised or self-supervised settings.
>
> For instance, Diffusion Models have been shown to suffer from dimensional collapse. Recently, Wang et al. [2] proposed to regularize a generative model's internal representations by encouraging their dispersion in the hidden space.
>
> In multi-modal applications, several papers [3,4] have highlighted that uniformity properties can be highly beneficial to bridge the *modality gap* [5] in multi-modal models. The *modality gap* is a phenomenon observed in pre-trained multimodal models, such as CLIP [6], which provide transferable embeddings mainly by aligning images and text in the same embedding space. Better uniformity allows for the embeddings from the different modalities to more evenly distribute on the hypersphere in the embedding space. To validate the generalizability, we assess our method on image-text retrieval, a representative vision-language task, following the same experimental setting as Oh et al. (Section 5.2) [3] on Flickr30k and MS-COCO. More precisely, we apply T-REG as a regularizer to finetune CLIP: ($\mathcal L = \mathcal L_{\text{CLIP}} + \mathcal L_{\text{T-REG}}$), where $\mathcal L_{\text{CLIP}}$ is the standard CLIP loss [6] and $\mathcal L_{\text{T-REG}}$ refers to Eq. (7). We report the top 1/5 Recall (R@1, R@5) in Table B.2 below with Image-to-text (i → t) and text-to-image (t → i) retrieval results. Except for the last line, the table is inherited from Oh et al. [3] (bold indicates the best performance). Complete results and implementation details will be included in the supplementary material of the paper.
>
> #### Table B.2: Image-to-text (i → t) and text-to-image (t → i) retrieval results
> | Method | Flickr30k  | | |  | | | | MS-COCO  | | | | |
> |--------|:---------------:|:---------:|---|:---:|----|---------|---|:-----:|:-----:|-----|:----:|:-----:|
> | | i→t |  |  |  t→i | | ||  i→t | ||  t→i |
> | | R@1 | R@5 | | R@1 | R@5 | | | R@1 | R@5 | | R@1 | R@5 |
> | Zero-Shot | 71.1 | 90.4 | | 68.5 | 88.9 | | | 31.9 | 56.9 | | 28.5 | 53.1 |
> | Finetune | 81.2 | 95.5 | | 80.7 | 95.8 | | | 36.7 | 63.6 | |  36.9 | 63.9 |
> | ES [7]| 71.8 | 90.0 | | 68.5 | 88.9 | | | 31.9 | 56.9 | |  28.7 | 53.0 |
> | i-Mix [8] | 72.3 | 91.7 | | 69.0 | 91.1 | | | 34.0 | 63.0 | |  34.6 | 62.2 |
> | Un-Mix [9] | 78.5 | 95.4 | | 74.1 | 91.8 | | | 38.8 | 66.2 | |  33.4 | 61.0 |
> | m³-Mix [5] | 82.3 | 95.9 | | **82.7** | 96.0 | | | 41.0 | 68.3 | |  39.9 | 67.9 |
> | **T-REG** | **83.2** | **96.0** | | 80.8 | **96.4** | | | **41.6** |**68.7**| | **41.5** | **68.7** |
>
> **Q3. “What is the extra cost of MST computation?”**
>
> We use Kruskal’s algorithm, which runs in $\mathcal{O}(B^2 (D + \log B))$ time in the worst case, where $B$ is the number of samples in the batch and $D$ is the dimension, with a resulting computational overhead that is similar to well-established methods such as VICReg or SimCLR; and faster wall-clock time per step.  We report the complexities in the table below, along with wall-clock time per step, following a suggestion by Reviewer MGro.
>
> #### Table A.1: Compute comparison. The ranges are reported from Garrido et al [4] for SimCLR and Bardes et al [3] for VICReg.
> | Method | Loss Complexity | B range | D range | Wall-clock time (per step, sec) |
> | - | - | - | - | :-:|
> | SimCLR | $\mathcal{O}(B^2⋅D)$ | [2048-4096] | [256-1024] | 0.22 +/- 0.03 |
> | VICReg | $\mathcal{O}(B⋅D^2)$ | [512-2048] | [4096-8192] | 0.23 +/- 0.02 |
> | T-REGS | $\mathcal{O}(B^2 (D + \log B))$ | [512-1024] | [512-2048] | 0.20 +/- 0.001 |
>
>
> ---
>
> References:
>
> [1] T. M. Apostol and M. A. Mnatsakanian: “New Horizons in Geometry”, Mathematical Association of America, Dolciani Mathematical Expositions #47, 2012, ISBN 978-0-88385-354-2.
>
> [2] R. Wang, K. He. Diffuse and Disperse: Image Generation with Representation Regularization. Arxiv 2025
>
> [3] C. Oh, J. So, H. Byun, Y. Lim, M. Shin, J-.J. Jeon, K. Song. Geodesic Multi-Modal Mixup for Robust Fine-Tuning. NeurIPS 2023
>
> [4] J. Kim, S. Hwang.   Enhanced OoD Detection through Cross-Modal Alignment of Multi-Modal Representations. In CVPR, 2025
>
> [5]  V. W. Liang, Y. Zhang, Y. Kwon, S. Yeung, and J. Y. Zou. Mind the gap: Understanding the modality gap in multi-modal contrastive representation learning. NeurIPS, 2022
>
> [6] A. Radford, J. W. Kim, C. Hallacy, A. Ramesh, G. Goh, S. Agarwal, G. Sastry, A. Askell,
> P. Mishkin, J. Clark, et al. Learning transferable visual models from natural language supervision.
> In Proc. ICML, 2021
>
> [7] A. C. Li, A. A. Efros, and D. Pathak. Understanding collapse in non-contrastive siamese
> representation learning. In ECCV, 2022
>
> [8] K. Lee, Y. Zhu, K. Sohn, C.-L. Li, J. Shin, and H. Lee. i-mix: A domain-agnostic strategy for contrastive representation learning. In ICLR, 2021.
>
> [9] Z. Shen, Z. Liu, Z. Liu, M. Savvides, T. Darrell, and E. Xing. Un-mix: Rethinking image
> mixtures for unsupervised visual representation learning.  In AAAI, 2022
>
> [10] Q. Garrido, Y. Chen, A. Bardes, L. Najman, and Y. Lecun. On the duality between contrastive and non-contrastive self-supervised learning. In ICLR, 2023
>
> [11] A. Bardes, J. Ponce, and Y. Lecun. Vicreg: Variance-invariance-covariance regularization for self-supervised learning. In ICLR, 2022.

---

> > ### Comment · Reviewer_hUvh · 2025-08-05
> > **Replying to Rebuttal**
> >
> > I appreciate the extra experiments and the improvement in theoretical analysis. I hope our discussion can be added to the final version of the paper. Based on the reasonable solutions to the questions I raised, I have decided to raise my score and recommend acceptance.

---

> > > ### Author Response · Authors · 2025-08-05
> > >
> > > We thank you for your feedback and for raising your score to recommend acceptance. We are currently adding the content of our discussion to the paper, mainly in Sections 4 and 5 (plus some sentences in the introduction and some supplementary material), so it will appear in the final version.

---

### Official Review · Reviewer_MGro · 2025-07-01

**Clarity:** 4
**Significance:** 3
**Originality:** 4
**Rating:** 5
**Confidence:** 4

**Summary:**

This paper introduces T-REGS, a novel regularization framework for joint-embedding self-supervised learning (JE-SSL). The core idea is to regularize the learned embeddings by maximizing the length of the Minimum Spanning Tree (MST) constructed over the embedding vectors within a batch, while simultaneously applying a soft constraint to keep the embeddings on the unit hypersphere.

The authors claim that this approach simultaneously addresses two critical issues in representation learning:

Dimensional Collapse: It encourages the embeddings to occupy the full dimensionality of the representation space.
Sample Uniformity: It promotes an even distribution of embeddings on the hypersphere.
The paper provides a solid theoretical analysis, connecting the MST length maximization to statistical dimension estimation and Rényi entropy maximization. This analysis shows that for small sample sizes, the optimal configuration of points under the T-REG loss forms a regular simplex, and asymptotically for large samples, it encourages a uniform distribution. The method's effectiveness is validated through experiments on synthetic data, which clearly illustrate its properties, and on standard SSL benchmarks (CIFAR-10/100, ImageNet-100/1k), where it achieves competitive performance compared to state-of-the-art methods.

**Questions:**

refer to weaknesses.

**Ethical Concerns:**

["NO or VERY MINOR ethics concerns only"]

**Final Justification:**

The authors not only address all problems from my comments, but also the authors make the proposition more convincing by extending the core proposition to a more relaxed assumption with extra experiments to improve existing results.
Hence, I keep my rate "accept".

**Limitations:**

refer to weaknesses.

**Paper Formatting Concerns:**

no format issues.

**Quality:**

3

**Strengths And Weaknesses:**

Strengths
1.Novelty and Elegance of the Core Idea: The central contribution—using the length of the Minimum Spanning Tree as a regularizer—is highly novel and conceptually elegant. It draws a clever and principled connection between a classic algorithm from graph theory, concepts from statistical dimension estimation, and the modern challenges of self-supervised representation learning. This is a significant conceptual contribution that could inspire new research directions.
2.Strong Theoretical Grounding: The paper is not just a heuristic proposal; it is backed by rigorous theoretical analysis (Section 4.1).
Proposition 4.1 provides a clear and intuitive result for the small-batch regime (n ≤ d + 1), showing that the loss is maximized when embeddings form a regular simplex. This directly proves that the regularizer prevents dimensional collapse in this setting.
Theorem 4.2 and Proposition 4.3 establish an asymptotic link between maximizing the MST length and maximizing the Rényi entropy of the embedding distribution. Since maximum entropy on a compact set corresponds to the uniform distribution, this provides a strong justification for the claim that T-REGS promotes sample uniformity.
3.Simplicity and Practicality: Compared to some competing methods that aim for similar goals, T-REGS is relatively simple. For instance, the Optimal Transport-based regularizer proposed by Fang et al. can be computationally expensive and numerically unstable. T-REGS, while requiring an MST computation, avoids complex operations like SVD and is conceptually straightforward to implement using existing fast, parallelized MST algorithms.

Weaknesses

1.Empirical Performance: While the results are competitive, T-REGS does not consistently set a new state-of-the-art on the reported benchmarks. For example, on CIFAR-100, it lags behind several W2-regularized methods, and on ImageNet-1k, it is slightly outperformed by methods like INTL and Zero-CL. While SOTA performance is not a strict requirement for a conceptually novel paper, a stronger showing would make the practical case for adoption more compelling. The authors acknowledge this limitation in the conclusion [305-306].
2.Computational Overhead of MST: The paper states that fast GPU-based MST implementations exist, but it does not provide a practical analysis of the computational overhead. For a batch of size B, constructing the complete graph has a complexity of O(B²), and standard MST algorithms like Kruskal's or Prim's are at least O(B²). This could become a bottleneck for very large batch sizes, which are common in SSL. A comparison of wall-clock training time per step against baselines like VICReg (which computes a covariance matrix, O(B*D²)) or BarlowTwins would be a valuable addition to assess the practical scalability.

---

> ### Author Rebuttal · Authors · 2025-07-29
>
> We thank you for your feedback and positive comments. Please find below our answers to the weaknesses you raised:
>
> **W1. “1.Empirical Performance: While the results are competitive, T-REGS does not consistently set a new state-of-the-art on the reported benchmarks. For example, on CIFAR-100, it lags behind several W2-regularized methods, and on ImageNet-1k, it is slightly outperformed by methods like INTL and Zero-CL. While SOTA performance is not a strict requirement for a conceptually novel paper, a stronger showing would make the practical case for adoption more compelling. The authors acknowledge this limitation in the conclusion [305-306].“**
>
> In order to strengthen the empirical validation of our regularization method, we followed suggestions by Reviewers Hgsb and hUvh and conducted further experiments to integrate T-REG as an auxiliary regularizer to BYOL and Barlow Twins on CIFAR-10/100. The results are reported in Table A.2 below, which highlights the benefits of such combinations (best performances are shown in bold):
>
> #### Table A.2: CIFAR-10/100 Evaluation
> | | CIFAR-10 |CIFAR-100 |
> |-|:-:|:-:|
> | MoCo v2 | 90.7 | 60.3 |
> | MoCo v2 + $\mathcal{L}_u$| 91.0 | 61.2 |
> | MoCo v2 + $\mathcal{W}_2$| 91.4 | 63.7 |
> | ZeroCL | 91.4 | **68.5** |
> | ZeroCL + $\mathcal{L}_u$ | 91.3 | 68.4 |
> | ZeroCL + $\mathcal{W}_2$ | 91.4 | **68.5** |
> | BYOL | 89.5 | 63.7 |
> | BYOL + $\mathcal{L}_u$ | 90.1 |62.7 |
> | BYOL + $\mathcal{W}_2$ | 90.1 |65.2 |
> | BYOL + **T-REG** | 90.4 | 65.7 |
> | Barlow Twins | 91.2 | 68.2 |
> | Barlow Twins + $\mathcal{L}_u$ | 91.4 | 68.4 |
> | Barlow Twins + $\mathcal{W}_2$ | 91.4 | **68.5** |
> | Barlow Twins + **T-REG** | **91.8** | **68.5** |
>
> We observe that adding our regularizer improves the results both for BYOL and Barlow Twins, highlighting its superiority.
>
> **W2  “2.Computational Overhead of MST: The paper states that fast GPU-based MST implementations exist, but it does not provide a practical analysis of the computational overhead. For a batch of size B, constructing the complete graph has a complexity of O(B²), and standard MST algorithms like Kruskal's or Prim's are at least O(B²). This could become a bottleneck for very large batch sizes, which are common in SSL. A comparison of wall-clock training time per step against baselines like VICReg (which computes a covariance matrix, O(BxD²)) or BarlowTwins would be a valuable addition to assess the practical scalability.”**
>
> Indeed, the worst-case asymptotic complexity is significant, but not larger than for other methods. Meanwhile, the hidden constants in the complexity bounds are small, which has an impact in practice since the batch sizes are relatively small (only a few thousand points). Furthermore, while Kruskal’s algorithm per se is sequential, its pre-processing (namely, computing the distance matrix and sorting its entries) is the most costly step. In practice, one can use a parallelized GPU implementation for it. This is what the torchPH library (used in our implementation) does.
>
> To confirm this insight, we evaluated the computational overhead of T-REGS, VICReg (a covariance-based method), and SimCLR (a contrastive method). We report the results in the table below, where $B$ represents the batch size and $D$ represents the output dimension. The wall-clock comparisons are averaged over 500 steps for each method.
>
> #### Table A.1: Compute comparison. The ranges are reported from Garrido et al [4] for SimCLR and Bardes et al [3] for VICReg.
> | Method | Loss Complexity | B range | D range | Wall-clock time (per step, sec) |
> | - | - | - | - | :-:|
> | SimCLR | $\mathcal{O}(B^2⋅D)$ | [2048-4096] | [256-1024] | 0.22 +/- 0.03 |
> | VICReg | $\mathcal{O}(B⋅D^2)$ | [512-2048] | [4096-8192] | 0.23 +/- 0.02 |
> | T-REGS | $\mathcal{O}(B^2 (D + \log B))$ | [512-1024] | [512-2048] | 0.20 +/- 0.001 |
>
> We observe a resulting computational overhead that is similar to well-established methods such as VICReg or SimCLR; and faster wall-clock time per step.
>
> ---
>
> References:
>
> [1] A. Bardes, J. Ponce, and Y. Lecun. Vicreg: Variance-invariance-covariance regularization for self-supervised learning. In ICLR, 2022.
>
> [2] Q. Garrido, Y. Chen, A. Bardes, L. Najman, and Y. Lecun. On the duality between contrastive and non-contrastive self-supervised learning. In ICLR, 2023

---

> > ### Comment · Reviewer_MGro · 2025-08-06
> > **Reply by Reviewer MGro**
> >
> > Thanks to the author's extra experiments and detailed feedback.
> > All my concerns are addressed. I will keep my rate.

---

> > > ### Author Response · Authors · 2025-08-08
> > >
> > > We thank you for maintaining your score and recommending acceptance.

---

### Official Review · Reviewer_zFs8 · 2025-07-02

**Clarity:** 2
**Significance:** 3
**Originality:** 3
**Rating:** 4
**Confidence:** 3

**Summary:**

To avoid dimensional collapse and enhance uniformity in self-supervised learning, the authors proposed the Minimum Spanning Tree (MST) over the learned representation. Moreover, the authors provided theoretical analysis.

**Questions:**

(1) In lines 66-70, the author introduced the concept of $\alpha$-minimum spanning tree, but the author didn’t combine it well with the context of self-supervised learning. I’m very curious about how to construct a connected acyclic graph given the embeddings of self-supervised learning.

(2) I want to know how the Minimum Spanning Tree solves the problems of avoiding dimensional collapse and enhancing uniformity in self-supervised learning.

**Ethical Concerns:**

["NO or VERY MINOR ethics concerns only"]

**Final Justification:**

In the initial submission, the author did not introduce the integration of Kruskal’s algorithm into the SSL pipeline, which made me unable to understand how the author implemented the MST algorithm in SSL. After the discussion with the authors, they clarified this pipeline. Therefore, I would like to raise my rating.

**Quality:**

3

**Strengths And Weaknesses:**

**Strengths:**

(1) The authors proposed the Minimum Spanning Tree (MST) to avoid dimensional collapse and enhance uniformity in self-supervised learning, which is a novel approach.

(2) The authors provided a great deal of theoretical analysis for the proposed Minimum Spanning Tree (MST).


**Weaknesses:**

(1) The authors did not provide a relatively clear explanation of how the Minimum Spanning Tree (MST) is combined with Self-Supervised learning. The picture provided by the authors in Figure 1 is common knowledge. Instead of providing a common-sense picture, the authors should have provided a picture in the introduction to help readers intuitively understand how the Minimum Spanning Tree (MST) is combined with Self-Supervised learning.

(2) I didn’t understand how the Minimum Spanning Tree solves the problems of avoiding dimensional collapse and enhancing uniformity in self-supervised learning.

I admit that I haven’t fully understood the author’s paper. If the author can address my two main concerns and some detailed issues, I will raise the score.

---

> ### Author Rebuttal · Authors · 2025-07-29
>
> We thank you for your feedback. Please find below our answers to the questions and weaknesses you raised, and let us know how we can help further clarify the method for you:
>
> **W1: “The authors did not provide a relatively clear explanation of how the Minimum Spanning Tree (MST) is combined with Self-Supervised learning. The picture provided by the authors in Figure 1 is common knowledge. Instead of providing a common-sense picture, the authors should have provided a picture in the introduction to help readers intuitively understand how the Minimum Spanning Tree (MST) is combined with Self-Supervised learning.”**
>
> We agree that Figure 1 in the introduction is standard, and that a more targeted illustration would help clarify how the MST is integrated into the SSL framework. We will modify the figure to better demonstrate  how the minimum spanning tree is constructed from embeddings and how it fits into the pipeline. For this, we will take inspiration from the visuals of Figure 2, which will be moved from Page 7 to the introduction to provide a better initial intuition for the reader.
>
> We will also add a few clarifying sentences to the introduction, in the same spirit as our answer to Q1 below.
>
>
> **Q1:  “In lines 66-70, the author introduced the concept of alpha-minimum spanning tree, but the author didn’t combine it well with the context of self-supervised learning. I’m very curious about how to construct a connected acyclic graph given the embeddings of self-supervised learning.”**
>
> We kindly refer you to Equation (10) in Section 5, which explains this point.
>
> Our regularization is applied separately on the embeddings of the two transformations of the batch obtained from the two network branches. Specifically, denoting by $Z=(z_1, \dots ,z_n)$ and $Z' = (z_1', \dots, z_n')$
> these two embeddings in $d$ dimensions, two minimum spanning trees $\mathrm{MST}(Z)$ and $\mathrm{MST}(Z’)$ are constructed on these two sets of points. The MST of $Z$ treats each point $z_i$ as a vertex, and connects the vertices by edges to form the connected acyclic graph of minimum total length, where the length of an edge corresponds to the Euclidean distance between its vertices. Likewise for the MST of $Z'$. The regularization terms in Equation (10) tend to maximize the total lengths of these two minimum spanning trees while not expanding the point clouds too much, which provably increases their entropy and dimensionality (please refer to our answer to W2 and Q2 below for the explanation).
>
> We intend to add one or two sentences in the same spirit as above to the introduction in order to clarify this point.
>
> **W2 & Q2 : “I didn’t understand how the Minimum Spanning Tree solves the problems of avoiding dimensional collapse and enhancing uniformity in self-supervised learning.”/  “I want to know how the Minimum Spanning Tree solves the problems of avoiding dimensional collapse and enhancing uniformity in self-supervised learning.”**
>
> The behavior of T-REG is explained by Proposition 4.1, which we have now extended to allow the points to lie inside the ball and not just on the sphere. First, minimizing the term $\mathcal{L}_E$ in Equation (7) expands the point cloud until the sphere constraint $\mathcal{L}_S$ becomes the dominating term; at that stage, the points stop expanding and start spreading themselves out uniformly along the sphere of directions, which prevents dimensional collapse and maximizes uniformity. The amount of expansion before spreading is prescribed by the strength of the sphere constraint term versus the $\mathrm{MST}$ term in the loss, which is driven by the ratio between their respective mixing parameters $\lambda$ and $\gamma$.
>
> For further visual intuition, we kindly refer you to the synthetic experiments in dimension 3 (Figures 2.a and 2.b, which will be moved to the introduction). We generate a highly concentrated point cloud of 256 points (orange points in Figures 2a and 2b). After optimization, T-REG successfully transforms this initially concentrated point cloud into a uniformly distributed point cloud on the sphere (blue points in Figure 2.a), while occupying the whole dimensional space—demonstrating no dimensional collapse. Without the sphere constraint, the optimization fails to converge as points diverge to infinity (Figure 2.b).

---

> ### Comment · Reviewer_zFs8 · 2025-08-01
> **Comments On Minimum Spanning Tree**
>
> While I appreciate the authors’ detailed responses, the critical issue concerning the integration of Minimum Spanning Trees (MST) with Self-Supervised Learning (SSL) remains inadequately addressed, both in the original manuscript and the rebuttal. I concur with Reviewer Hgsb regarding the lack of clarity.
>
> Major Concern (W1 & Q1: MST-SSL Integration): The fundamental mechanism of how MST is applied within the SSL framework is still unclear. The authors’ responses to my concerns (W1, W2, Q1), particularly W1 and Q1 regarding the core concept and rationale of MST usage, did not resolve the ambiguity. This lack of intuitive understanding makes it difficult for the SSL community to grasp the novelty and contribution of using MST in this context.
>
> It is essential that the revised manuscript provides a significantly clearer explanation. I strongly suggest supplementing the textual description with an intuitive visual illustration (e.g., diagram, flowchart) explicitly showing how the MST computation and structure are utilized within the SSL pipeline. Clarity on this point is paramount for acceptance.
>
> Pending substantial improvement on this central issue, I cannot currently recommend acceptance. I maintain my score.

---

> > ### Author Response · Authors · 2025-08-02
> > **Re: Comments on Minimum Spanning Tree**
> >
> > We thank you for your prompt response. It would help us address your concern if you could narrow it down somewhat, and make more explicit requests for change. Here are a few specific aspects that you might have in mind:
> > 1. What is an MST? We defined the MST to be a connected acyclic graph on the points, perhaps this terminology from graph theory is unfamiliar to some in SSL and would deserve to be elaborated on in your view, in which case we can provide a formal definition in the paper.
> > 2. How is the MST constructed from a point cloud (here a batch of embeddings)? There are several algorithms to do so, among which Kruskal's algorithm is standard and used by our implementation. We can outline it in the paper if you believe it is important to do so.
> > 3. Where and how is the MST computation inserted in the SSL pipeline? This is the topic of our response to your W1 and Q1. If it is still not clear to you, then please let us know which sentences precisely are unclear, so we can further elaborate on them.
> > 4. Illustration for point 3: we understand that you request for Figure 1 to be changed, and we agree that it will be beneficial to make it more specific to our approach. Unfortunately we cannot show you the result, because we cannot attach images to our rebuttal or comments. We explained verbally how we intend to change it, in response to your W1. Please tell us if you require further details in this description.
> > 5. How does maximizing the length of the MST help prevent dimensional collapse and promote sample uniformity? This is the topic of our response to your W2 and Q2. If it is still not clear to you, then please let us know which sentences precisely are unclear, so we can further elaborate on them. Also, please check our response to W1 of Reviewer hUvh, which gives further insight into the guarantees provided by Proposition 4.1 and its extension. This insight we intend to add in Section 4 of paper.
> > 6. Any other aspect? Please let us know precisely, and we will try to clarify it for you.

---

> > > ### Comment · Reviewer_zFs8 · 2025-08-07
> > > **Further Discussion With Authors**
> > >
> > > My primary concern initially stems from the question: “How is the MST constructed from a point cloud (in this case, a batch of embeddings)? There are multiple algorithms available for this purpose, and Kruskal’s algorithm, a standard one, is employed in our implementation.” Would you be willing to elaborate further on how to utilize Kruskal’s algorithm to establish a connection between MST and SSL?

---

> > > > ### Author Response · Authors · 2025-08-07
> > > >
> > > > We thank you for clarifying your concern.
> > > >
> > > > The main insight is that embeddings in SSL can be viewed as points in a high-dimensional space, forming a point cloud. Given this point cloud, we can compute all pairwise distances between embeddings. We then build the minimum spanning tree (MST) using Kruskal’s algorithm.
> > > > Here are the details of the integration of Kruskal's algorithm into the SSL pipeline:
> > > >
> > > > 1.  **Compute embeddings:** Compute embeddings $Z=(z_1, ..., z_n)$ using one branch of the SSL architecture described in Fig. 1. This forms a point cloud in $\mathbb{R}^d$. Likewise, compute embeddings $Z'=(z_1', ..., z_n')$ using the other branch of the SSL architecture, forming another point cloud in $\mathbb{R}^d$.
> > > > 2.  **Construct MST for $Z$**: From the pairwise Euclidean distances between the points $z_1,...,z_n$, use Kruskal's algorithm to build a tree $G$ of minimum total edge length that connects all of the points of $Z$ together. Kruskal's algorithm builds $G$ iteratively, starting with $G$ being the scattered set of nodes $z_i$ and ending with $G$ being a spanning tree for $Z$.
> > > > The construction considers all possible pairs $(z_i, z_j)$, by increasing order of distance: for each pair, it adds a corresponding edge in $G$ if that edge connects previously disconnected components of $G$. An exchange argument shows that this construction produces a spanning tree of minimum total length.
> > > > 3. **Construct MST for $Z'$:** Apply the same construction for $Z'$, which builds a second MST $G'$ with minimal total length containing the vertices $Z'$.
> > > > 4. **Compute the loss:** $\mathcal{L} = \mathcal{L}_ \text{MSE} (Z,Z') + \mathcal{L}_ \text{T-REG} (Z) + \mathcal{L}_ \text{T-REG} (Z')$ from (Eq. 10).
> > > > 5. **Perform an optimizer step**:  For each MST term in the loss, this means applying the formula of Eq. (4), which, in essence, makes each pair of points $(z_i, z_j)$ forming an edge in the MST repel each other.

---

> > > > > ### Comment · Reviewer_zFs8 · 2025-08-08
> > > > > **Further Discussion With Authors**
> > > > >
> > > > > Thank you very much to the author for the clarification on the integration of Kruskal’s algorithm into the SSL pipeline. I believe this pipeline is of great importance. I suggest that the author should include a separate section to introduce this pipeline so that readers in the SSL community can have a better understanding. I will raise the score to 4.

---

> > > > > > ### Author Response · Authors · 2025-08-08
> > > > > >
> > > > > > Thank you for your positive feedback and for raising your score to 4. We will use our discussion to complete the beginning of Section 5 with a paragraph clarifying the integration of the MST with the SSL pipeline; update Figure 1; and revise Algorithm 1 in the Appendix.

---

### Official Review · Reviewer_Hgsb · 2025-07-03

**Clarity:** 2
**Significance:** 2
**Originality:** 2
**Rating:** 4
**Confidence:** 4

**Summary:**

This paper proposes T-REGS as a regularization technique to SSL method which encourages uniformity of the representation. T-REGS maximizes the length of the minimum spanning tree, which connects to entropy maximization.

**Questions:**

1. Could you elaborate how exactly the length of minimum spanning tree is computed? Which algorithm is used for finding the minimum spanning tree found given a batch of features and what's the complexity of the algorithm? Could you compare the complexity with other regularization techniques?

2. It's unclear to me why T-REGS only uses MSE as the representation learning loss. Why would it not be able to act as a generic regularization for all SSL methods?

3. Do you have any ablations for the value of alpha?

4. In many popular SSL framework, the features are normalized to unit sphere. It has also been shown such normalization is beneficial. Why does the paper propose to regularize it via the soft sphere-constraint instead of a hard constraint? Could you provide empirical evidence suggesting this way better?

**Ethical Concerns:**

["NO or VERY MINOR ethics concerns only"]

**Final Justification:**

I am satisfied with the clarification during rebuttal and modifications to the paper the authors promised.

**Limitations:**

Yes

**Quality:**

2

**Strengths And Weaknesses:**

**Strength**

The theoretical foundation seems to be sound.

**Weakness**

The writing of the paper can be greatly improved. E.g. proofs can be moved to appendix. In the main paper, instead of showing proofs, explain why these theoretical results are important for its application here.

Comparing to other existing regularization techniques, T-REGS doesn't seem to result in better performance. It's unclear what's the benefit of adopting T-REGS compared to other methods.

---

> ### Author Rebuttal · Authors · 2025-07-29
>
> We thank you for your feedback. Please find below our answers to the questions and weaknesses you raised:
>
> **W1: “The writing of the paper can be greatly improved. E.g., proofs can be moved to the appendix. In the main paper, instead of showing proofs, explain why these theoretical results are important for its application here.”**
>
> We chose to include the proofs of the two pivotal results (Proposition 4.1, Proposition 4.3) in the main text because these results are central to the paper and explain the behavior of T-REG, with relatively short and elementary proofs. The final version of the paper will provide intuitive explanations of the role of the MST for preventing dimensional collapse and promoting well-distributed embeddings, based on Propositions 4.1 and 4.3—see our answer to Weakness 2 of Reviewer zFs8 for the details. This will add a few sentences to the introduction and Section 4, which will fit together with the new experimental results within the extra allocated page.
>
> **W2. “Comparing to other existing regularization techniques, T-REGS doesn't seem to result in better performance. It's unclear what's the benefit of adopting T-REGS is compared to other methods.**
>
> The main benefits of T-REGS stem from its conceptual and implementation simplicity, strong theoretical grounding, and robustness to hyperparameter variations (i.e., batch size and embedding dimension). Indeed, T-REGS offers a novel, conceptually simple solution based on well-understood graph theory and statistical dimension estimation. It follows the principled line of thought of Fang et al. [1] while proposing a more straightforward and unexplored approach: maximizing the length of the minimum spanning tree, which we found to satisfy the same properties naturally but with significantly lower computational overhead in practice and improved numerical stability. As a result, T-REGS demonstrates better scalability on larger datasets such as ImageNet-100 and ImageNet-1k.
>
> Compared with other existing approaches, T-REGS addresses the following well-known limitations:
>
> *(i) Theoretical grounding*: Asymmetric methods lack theoretical justification to explain how the architecture helps prevent collapse [2]. Covariance-based methods only leverage the second moment of the data distribution and are thus blind to the concentration points of the density, which can prevent convergence to the uniform distribution (Figure 5 in the Supp. Mat.),
>
> *(ii) Computational limitations*: contrastive methods are sensitive to the number of negative samples and require large batch sizes [3, 4]. Covariance-based approaches often rely on both large batch sizes and large output dimensions [2].  By contrast, T-REGS proves to be robust to both batch size and embedding dimensions (Appendices D.1 and D.2).
>
> To strengthen the empirical validation of our method, we have conducted complementary experiments based on your suggestions. We propose a revised version of Table 1 that highlights the new results (please refer to our answer to Q2).
>
> **Q1: “Could you elaborate how exactly the length of minimum spanning tree is computed? Which algorithm is used for finding the minimum spanning tree found given a batch of features and what's the complexity of the algorithm? Could you compare the complexity with other regularization techniques?”**
>
> In practice, we use Kruskal’s algorithm, which runs in $\mathcal{O}(B^2 (D + \log B))$ time in the worst case, where $B$ is the number of samples in the batch and $D$ is the dimension. This choice was motivated by the simplicity of its implementation, with a resulting computational overhead that is similar to well-established methods such as VICReg or SimCLR; and faster wall-clock time per step. We report the complexities in Table A.1 below, along with wall-clock time per step, following a suggestion by Reviewer MGro.
>
> #### Table A.1: Compute comparison. The ranges are reported from Garrido et al [4] for SimCLR and Bardes et al [3] for VICReg.
> | Method | Loss Complexity | B range | D range | Wall-clock time (per step, sec) |
> | - | - | - | - | :-:|
> | SimCLR | $\mathcal{O}(B^2⋅D)$ | [2048-4096] | [256-1024] | 0.22 +/- 0.03 |
> | VICReg | $\mathcal{O}(B⋅D^2)$ | [512-2048] | [4096-8192] | 0.23 +/- 0.02 |
> | T-REGS | $\mathcal{O}(B^2 (D + \log B))$ | [512-1024] | [512-2048] | 0.20 +/- 0.001 |
>
>
> **Q2: “It's unclear to me why T-REGS only uses MSE as the representation learning loss. Why would it not be able to act as a generic regularization for all SSL methods?”**
>
> T-REG is indeed designed as a generic regularization technique and, in principle, can be combined with a wide range of self-supervised learning (SSL) objectives—not just as a standalone regularization with an MSE. In the submitted version of the paper, we decided to emphasize the behavior of T-REGS (T-REG combined with MSE invariance) as a standalone method. Given the feedback received, we have run further experiments combining T-REG with a couple of other regularizers: BYOL and Barlow Twins, on CIFAR-10/100. We provide the corresponding results below in Table A.2, which highlight the benefits of such combinations (best performances are shown in bold):
>
> #### Table A.2: CIFAR-10/100 Evaluation
> | | CIFAR-10 |CIFAR-100 |
> |-|:-:|:-:|
> | MoCo v2 | 90.7 | 60.3 |
> | MoCo v2 + $\mathcal{L}_u$| 91.0 | 61.2 |
> | MoCo v2 + $\mathcal{W}_2$| 91.4 | 63.7 |
> | ZeroCL | 91.4 | **68.5** |
> | ZeroCL + $\mathcal{L}_u$ | 91.3 | 68.4 |
> | ZeroCL + $\mathcal{W}_2$ | 91.4 | **68.5** |
> | BYOL | 89.5 | 63.7 |
> | BYOL + $\mathcal{L}_u$ | 90.1 |62.7 |
> | BYOL + $\mathcal{W}_2$ | 90.1 |65.2 |
> | BYOL + **T-REG** | 90.4 | 65.7 |
> | Barlow Twins | 91.2 | 68.2 |
> | Barlow Twins + $\mathcal{L}_u$ | 91.4 | 68.4 |
> | Barlow Twins + $\mathcal{W}_2$ | 91.4 | **68.5** |
> | Barlow Twins + **T-REG** | **91.8** | **68.5** |
>
> We observe that adding our regularizer improves the results both for BYOL and Barlow Twins, highlighting its superiority.
>
> **Q3:  “Do you have any ablations for the value of alpha?”**
>
> We tested  $\alpha=1$ and $\alpha=0.5$ and found that $\alpha=1$ works best; however, we have not performed an exhaustive hyperparameter sweep. Table A.3 reports the results:
>
> #### Table A.3: Ablation of $\alpha$. Results on CIFAR-10/CIFAR-100.
> | $\alpha$ | CIFAR-10 | CIFAR-100|
> |--|:--:|:--:|
> |$\frac{1}{2}$ | 90.6 | 66.0 |
> | 1 | 91.2 | 66.8 |
>
>
> **Q4: “In many popular SSL framework, the features are normalized to unit sphere. It has also been shown such normalization is beneficial. Why does the paper propose to regularize it via the soft sphere-constraint instead of a hard constraint? Could you provide empirical evidence suggesting this way better?”**
>
> Indeed, some popular SSL frameworks such as SimCLR and BYOL employ explicit normalization of features to the unit sphere, enforcing a hard constraint. Others do not enforce such constraints explicitly, and instead rely on soft mechanisms—such as VICReg, which incorporates variance and covariance regularization terms in its loss, implicitly constraining the distribution (and to some extent, the norms) of the embeddings.
>
> We chose a soft sphere constraint for the following reasons:
> -  A hard normalization has been shown to ignore the importance of the embedding norm for gradient computation, whereas a soft constraint enables better embedding optimization [5,6]
> - Furthermore, during our initial experiment, we found that relaxing the sphere constraint from a hard one to a soft one provides more leeway for optimization and leads to improved results. Table A.4 shows the results with/without normalization.
>
> #### Table A.4: Results with and without normalization
> | | CIFAR-10 | CIFAR-100 |
> | - | :-: | :-: |
> | soft constraint | 91.2 | 66.8 |
> | hard constraint | 89.2 | 64.7 |
>
>
> ---
>
> References:
>
> [1] X. Fang, J. Li, Q. Sun, and B. Wang. Rethinking the uniformity metric in self-supervised learning. In ICLR, 2024
>
> [2] X. Weng, Y. Ni, T. Song, J. Luo, R. M. Anwer, S. Khan, F. Khan, and L. Huang. Modulate your spectrum in self-supervised learning. In ICLR, 2024
>
> [3] Q. Garrido, Y. Chen, A. Bardes, L. Najman, and Y. Lecun. On the duality between contrastive and non-contrastive self-supervised learning. In ICLR, 2023
>
> [4] A. Bardes, J. Ponce, and Y. Lecun. Vicreg: Variance-invariance-covariance regularization for self-supervised learning. In ICLR, 2022.
>
> [5] D. Zhang, Y. Li1, Z. Zhang. Deep Metric Learning with Spherical Embedding. NeurIPS 2020
>
> [6] J. Zhang, H. Zhang, R. Vasudevan, M. Johnson-Roberson. Hyperspherical Embedding for Point Cloud Completion. In CVPR 2023.

---

> > ### Comment · Reviewer_Hgsb · 2025-08-06
> >
> > Thank you for proving discussion of the complexity of the algorithm, more thorough comparison with baselines and ablations. In particular, it's great to see an improvement when combining T-REGS with other standard SSL methods on CIFAR10/100. However, I think the authors should add a discussion here as (some of) baselines don't seem to operate under their optimal regime in terms of hidden dimension (or training length).
> >
> > While I understand theoretical results are important for this paper, I still think the technicality of the proof can be moved to appendix, so that the main paper flows better and leaving space for more important discussions (such as complexity and empirical comparision). In addition, it would be much more convincing if similar comparison to the baseline is carried out on ImageNet1k. Subject to these improvements, I can raise my score to 4.

---

> > > ### Author Response · Authors · 2025-08-07
> > >
> > > We thank you for your answer.
> > >
> > > On moving the technical details of the proof to the appendix: The proof of our new Proposition 4.1 (please see our answer to W1 of Reviewer hUvh for the details) allows us to defer the technicalities to the appendix while keeping enough details in Section 4.1.1 for the interested reader. Here is the new structure of Section 4.1.1:
> > > - First, the statement of the new version of Proposition 4.1;
> > > - Then, the statements of the two main proof ingredients: Theorem 14.6 from [1] (only Eq. (14.22) is needed), and a lemma relating the total length of the MST to the total sum of pairwise distances.  The proof of the lemma, which is the main technical component here, is moved to the appendix.
> > > - Finally, a 1-line calculation combining these two ingredients to prove Proposition 4.1.
> > >
> > > Overall, the new Section 4.1.1 takes about 15 lines (versus 45 lines previously). If needed, the two main proof ingredients can be mentioned informally in the section, and their formal statements can be deferred to the appendix.
> > >
> > > On adding the complexity of building the MST: as this point was raised by several reviewers-- and we agree this is a critical consideration for larger models and datasets-- we will use the extra space to address these concerns.  Specifically, we will provide a ~10-line paragraph with empirical and theoretical details on MST complexity and compare it with other regularizers.  The remaining space will be used to provide more comprehensive details and a global figure showcasing the addition of MSTs for SSL, as suggested by Reviewer zFs8.
> > >
> > >
> > > As for the CIFAR10/100 experiments, we focused on a direct comparison with Fang et al. [2], the most relevant baseline for T-REG. Accordingly, we reported baseline results from their paper for consistency. We will clarify this in the final version.
> > >
> > > Finally, regarding experiments combining standard SSL methods with T-REG on ImageNet-1k: We are currently running experiments on ImageNet-1k; however, as these experiments are quite time-consuming, we will prioritize providing results on ImageNet-100. We already have the following result on ImageNet-100: Barlow Twins + T-REG achieves 80.9, compared to 80.2 for Barlow Twins alone.
> > >
> > > We will include the full set of results in the final version of the paper.
> > >
> > >
> > >
> > > ---
> > >
> > > References:
> > >
> > > [1] T. M. Apostol and M. A. Mnatsakanian: “New Horizons in Geometry”, Mathematical Association of America, Dolciani Mathematical Expositions #47, 2012, ISBN 978-0-88385-354-2.
> > >
> > > [2] X. Fang, J. Li, Q. Sun, and B. Wang. Rethinking the uniformity metric in self-supervised learning. In ICLR, 2024

---

> > > > ### Author Response · Authors · 2025-08-08
> > > >
> > > > As a follow-up to our previous message, you can find below in Table A.5 some additional results when combining T-REG with existing methods on ImageNet-100 (due to time constraints, we couldn't experiment with ImageNet-1k).
> > > >
> > > > ####  Table A.5: ImageNet-100 Complementary Evaluation
> > > >
> > > > | | ImageNet-100|
> > > > |--|:--:|
> > > > |Barlow Twins| 80.2 |
> > > > |Barlow Twins + **T-REG**| 80.9 |
> > > > |BYOL | 80.3 |
> > > > |BYOL + **T-REG** | 80.8 |
> > > >
> > > > We will include the full set of results in the final version of the paper.

---

> > > > > ### Comment · Reviewer_Hgsb · 2025-08-08
> > > > >
> > > > > Thanks for the additional results and the modifications to the paper. I raised the score to 4.

---

> > > > > > ### Author Response · Authors · 2025-08-08
> > > > > >
> > > > > > Thank you for your feedback and for raising your score.

---

### Note · Authors · 2025-08-12

We thank all reviewers for their insightful comments and the AC for their time. Reviewers agreed on several key strengths: the introduction of an MST-based regularizer for SSL is both novel and theoretically well-grounded; the approach is conceptually simple and easy to implement; T-REGS provides competitive results as a standalone regularizer, and new experimental results suggest good complementarity with existing ones.

In response to reviewers’ input, we made the following revisions:

- **Clarity** (R-Hgsb and R-zFs8): added a paragraph at the beginning of Section 5 to more clearly explain MST integration within the SSL pipeline; revised Fig. 1; and reorganized Section 4.1.1, moving technical details to the appendix while retaining essential information.
- **Theoretical guarantees** (R-hUvh):  extended Proposition 4.1 to incorporate the soft-sphere constraint, clarifying the regularizer’s behavior and reinforcing theoretical guarantees.
- **Empirical validation** (R-Hgsb, R-hUvh, and R-MGro): demonstrated consistent gains when combined with other regularizers (Rebuttal to R-Hgsb, Table A.2; comments to R-Hgsb, Table A.5); and conducted an additional experiment on a new task (Rebuttal to R-hUvh, Table B.2)
- **Computational complexity** (R-Hgsb, R-MGro, and R-hUvh): compared T-REGS to VICReg and SimCLR, finding similar theoretical complexity and practical computational overhead, while achieving faster wall-clock time per step (Rebuttal to Reviewer Hgsb, Table A.1).
- **Ablation studies**: on $\alpha$ (Rebuttal to Reviewer Hgsb, Table A.3), hard vs soft normalization (Rebuttal to Reviewer Hgsb, Table A.4), and additional loss coefficients (Rebuttal to Reviewer hUvh, Table B.1).

All reviewers acknowledged the improvements, with each either increasing their scores or maintaining positive scores. Thanks to the reviewers’ feedback, we believe the revisions strengthen our paper, demonstrating T-REGS' robustness and broader applicability while maintaining conceptual simplicity and theoretical foundations.

---

### Decision · Program_Chairs · 2025-09-17

**Decision:**

Accept (spotlight)

**Comment:**

The manuscript provides a simple regularization framework for self-supervised learning based on the length of the Minimum Spanning Tree (MST) over the learned representation. The idea is simple and regularization-based, which means that its adaptability to existing methods is high. The idea is supported by a clear theoretical background. Although the original manuscript did not successfully convey the importances of the main theorems, they have been clarified in the author-reviewer discussions. It would not be difficult to include those explanations with the reviewers about the significance of those theorems. Other weaknesses, e.g., about empirical results and computational complexity, have been addressed in the discussion.